# Diverse silent chromatin states modulate genome compartmentalization and loop extrusion barriers

George Spracklin [1,2,7] ✉, Nezar Abdennur[1,7], Maxim Imakaev [1], Neil Chowdhury [3], Sriharsa Pradhan [4], Leonid A. Mirny [1,5] & Job Dekker [2,6] ✉

The relationships between chromosomal compartmentalization, chromatin state and function are poorly understood. Here by profiling long-range contact frequencies in HCT116 colon cancer cells, we distinguish three silent chromatin states, comprising two types of heterochromatin and a state enriched for H3K9me2 and H2A.Z that exhibits neutral three-dimensional interaction preferences and which, to our knowledge, has not previously been characterized. We find that heterochromatin marked by H3K9me3, HP1α and HP1β correlates with strong compartmentalization. We demonstrate that disruption of DNA methyltransferase activity greatly remodels genome compartmentalization whereby domains lose H3K9me3-HP1α/β binding and acquire the neutrally interacting state while retaining late replication timing. Furthermore, we show that H3K9me3-HP1α/β heterochromatin is permissive to loop extrusion by cohesin but refractory to CTCF binding. Together, our work reveals a dynamic structural and organizational diversity of the silent portion of the genome and establishes connections between the regulation of chromatin state and chromosome organization, including an interplay between DNA methylation, compartmentalization and loop extrusion.

Chromosome organization within the nucleus is associated with vital cellular processes[1–3]. The best characterized chromosome-organizing process is loop extrusion. During interphase, cohesin complexes act as motors to extrude progressively growing chromatin loops. In vertebrates, the insulator protein CTCF serves as a directional barrier that halts loop-extruding cohesin[4–9]. Independent of loop extrusion, chromosomes are also spatially compartmentalized, with transcriptionally active chromatin located centrally and inactive chromatin more peripherally in the nucleus. As independent organizing processes, perturbing loop extrusion and its barriers does not eliminate compartmentalization[10–14]; however, the two processes act simultaneously and therefore can interfere with each other[12,15].

Simulations of chromosome compartmentalization in inverted nuclei have suggested that attraction between heterochromatic loci is a major force driving compartmentalization[16]. Heterochromatin is usually categorized into two types. Facultative heterochromatin, which is considered to be developmentally regulated, is enriched in H3K27me3 (ref. 17), while constitutive heterochromatin is viewed

[1]Institute for Medical Engineering and Sciences, Massachusetts Institute of Technology (MIT), Cambridge, MA, USA. [2]Department of Systems Biology, University of Massachusetts Medical School, Worcester, MA, USA. [3]Program for Research in Mathematics, Engineering and Science for High School Students (PRIMES), MIT, Cambridge, MA, USA. [4]Genome Biology Division, New England Biolabs, Inc., Ipswich, MA, USA. [5]Department of Physics, Massachusetts Institute of Technology (MIT), Cambridge, MA, USA. [6]Howard Hughes Medical Institute, Chevy Chase, MD, USA. [7]These authors contributed equally: George Spracklin, Nezar Abdennur. ✉e-mail: george.spracklin@gmail.com; job.dekker@umassmed.edu

as more static, is primarily associated with H3K9me3 and forms at centromeres, pericentromeric regions and at telomeres[18]. However, H3K9me3-associated heterochromatin is also found to form large contiguous domains genome-wide that expand in number and size during differentiation from pluripotency[19]. HP1 proteins bind H3K9me3 (reviewed in ref. 20) and can self-oligomerize and recruit H3K9 methyltransferases potentially contributing to heterochromatin compaction[21,22], spread[23,24] and phase separation[25–27].

DNA methylation is associated with both heterochromatin and extrusion barriers. In humans, the DNA methyltransferase DNMT1 physically associates with HP1 proteins suggesting an interplay between DNA and histone methylation[28,29]. CTCF-DNA binding also depends on CpG methylation of the core binding motif[30–33]. Overall, the regulatory relationships between DNA methylation, CTCF binding and heterochromatin formation are likely critical for cell-type specification but are still poorly understood.

Early studies subdivided mammalian genomes on the basis of long-range contact frequencies into two groups or 'compartments', broadly correlating with active and inactive chromatin[34,35]. Higher resolution Hi-C data have shown that this binary classification is too simplistic. Until recently, most of these studies have largely focused on a single deeply sequenced immortalized lymphoid cell line, GM12878 (ref. 4). However, since the Hi-C profile of a single locus depends on the chromatin state of the remainder of the genome, long-range patterns can be difficult to generalize and compare across cell types. Conversely, even when congruences are found where a group of loci share similar interaction profiles in each of two different cell types, there is no guarantee that the underlying chromatin states are identical.

Here, we report a detailed investigation of nuclear compartmentalization motivated by the prominent compartmentalization of heterochromatin in HCT116 colon cancer cells. We identify three inactive chromatin states having coherent long-range contact profiles, including a state marked by H3K9me2 and the histone variant H2A.Z, which, to our knowledge, has not previously been characterized. We find a strong compartmentalization signature for heterochromatin marked by H3K9me3, HP1α and HP1β and demonstrate that this heterochromatin is lost upon DNA methylation inhibition to yield the H3K9me2-enriched state, dramatically altering genome compartmentalization but not replication timing. Finally, we reveal an interplay between heterochromatin and loop extrusion. Together, our results demonstrate diversity and plasticity in silent chromatin, and their influence on the two major chromosome-organizing processes in interphase.

## Results

### Identifying interaction profiles by spectral decomposition

Evidence exists that some cell lines or cell types may have unique nuclear compartmentalization and that this may be linked to the structural differences of distinct states of chromatin[36–38]. To this end, we sought to identify groups of loci with similar long-range three-dimensional (3D) interaction profiles in HCT116 cells and to understand their relationship to the chromatin landscape (Fig. 1a). Our method for characterizing interaction profiles leverages the information from *trans* (interchromosomal) interactions as in ref. 4 but introduces an initial dimensionality reduction step similar to ref. 39. Rather than clustering columns of Hi-C contact matrices directly, we replace the contact frequency data of individual loci with their dimensionally reduced representation (that is, leading eigenvectors; Methods). This representation also facilitates the projection and embedding of genomic loci to allow investigation of the structure of the interaction profile manifold, in which each point corresponds to a 50-kilobase (kb) genomic bin (Fig. 1b).

In contrast to the discrete compartment model, we observe that the manifold does not form dense, strongly separated clusters as evidenced by the relatively continuous uniform manifold

approximation and projection (UMAP) embedding of the leading eigenvectors (Fig. 1b and Methods). Furthermore, projecting loci onto the first two eigenvectors (E1 and E2), we notice that GC content and genomic distance from centromere of individual loci vary along almost perpendicular components in the projection (Fig. 1c). A similar pattern is observed in other cell types, suggesting that these two roughly independent gradients are conserved features (Extended Data Fig. 1a). The alignment of GC content to E1 is well known, but the exact relationship differs across cell types[35]. The positional component correlating strongly with E2 reflects the observation that pairs of centromere-proximal and centromere-distal regions show mildly elevated contact frequency throughout the genome (Fig. 1c)[35]. This may be due to known enrichment of interactions between telomeres and/or between centromeres (for example, Rabl configuration), or a relationship between chromosomal and nuclear landmarks during interphase. As a result, we expected that the clustering of interaction profiles using *trans* Hi-C data would be influenced by chromosomal position independently of chromatin state. To test this idea, we examined subcompartment calls from GM12878 (ref. 4). Indeed, the loci from inactive subcompartments B2 and B3 in GM12878 appear to differ positionally along the E2 axis (Extended Data Fig. 2a–e). Similarly, in HCT116 cells we observe that several pairs of clusters with similar E1 ranges separate along the E2 axis (Fig. 1b).

We found that the data can be sensibly partitioned into eight clusters (Methods and Extended Data Fig. 1d,e). To exclude the influence of genomic position, we next examined data obtained with functional genomic assays including publicly available data (Supplementary Table 1)[40,41]. Indeed, several centromere-proximal and distal pairs of clusters showed similar functional profiles, so we consolidated the clusters into a total of five groups, described in detail below. Importantly, since not all interaction profiles imply the existence of spatially or phase-separated subnuclear compartments (see below), we will refer to our consolidated classification as interaction profile groups (IPGs) rather than (sub)-compartments. For simplicity, we have chosen a naming system similar to the one used for GM12878 *trans* interaction profile clusters (subcompartments), but below we discuss what correspondences can be made.

We identified two transcriptionally active IPGs, consistent with previous reports[4]. The first IPG, corresponding to cluster I, has the strongest self-interaction preference in *trans*, is enriched for the nuclear speckle marker SON and displays the greatest amount of transcriptional activity (Fig. 1d–f). Its loci have a high degree of overlap with the A1 subcompartment identified in GM12878 cells and thus we termed this IPG $A_1$ (Extended Data Fig. 1b). In GM12878, subcompartment A2 has been described in more generic terms as domains with weak transcriptional activity. Thus, clusters II and III which display weak transcriptional activity and separate along the E2 axis were grouped and classified as $A_2$ (Fig. 1b,e). Interestingly, the $A_2$ IPG interacts with the $A_1$ IPG (heterotypic) at least as strongly as it does with itself (homotypic) (Fig. 1f).

The five remaining clusters all display low transcriptional activity and gene density and thus likely constitute inactive chromatin domains (Fig. 1e). Clusters V and VI are both enriched in LaminB1, are late replicating and have intermediate CpG methylation, consistent with the B1 subcompartment label, so we combined them to form an IPG termed $B_1$ (Fig. 1d). Clusters VII and VIII are both enriched in Protect-seq signal, are late replicating, display the lowest CpG methylation frequency (~50% on average, corresponding to partially methylated domains[42–44]) and have the strongest preference for homotypic contacts in *cis* (Fig. 1d,f). The majority of loci in these clusters are assigned subcompartment labels B2 and B3 in GM12878 cells and are consistently assigned labels B2/B3 across different cell types based on SNIPER (subcompartment inference using imputed probabilistic expressions)[36], a supervised model that generalizes the GM12878 labels to other cell types (Fig. 1g and Extended Data Fig. 1b). However,

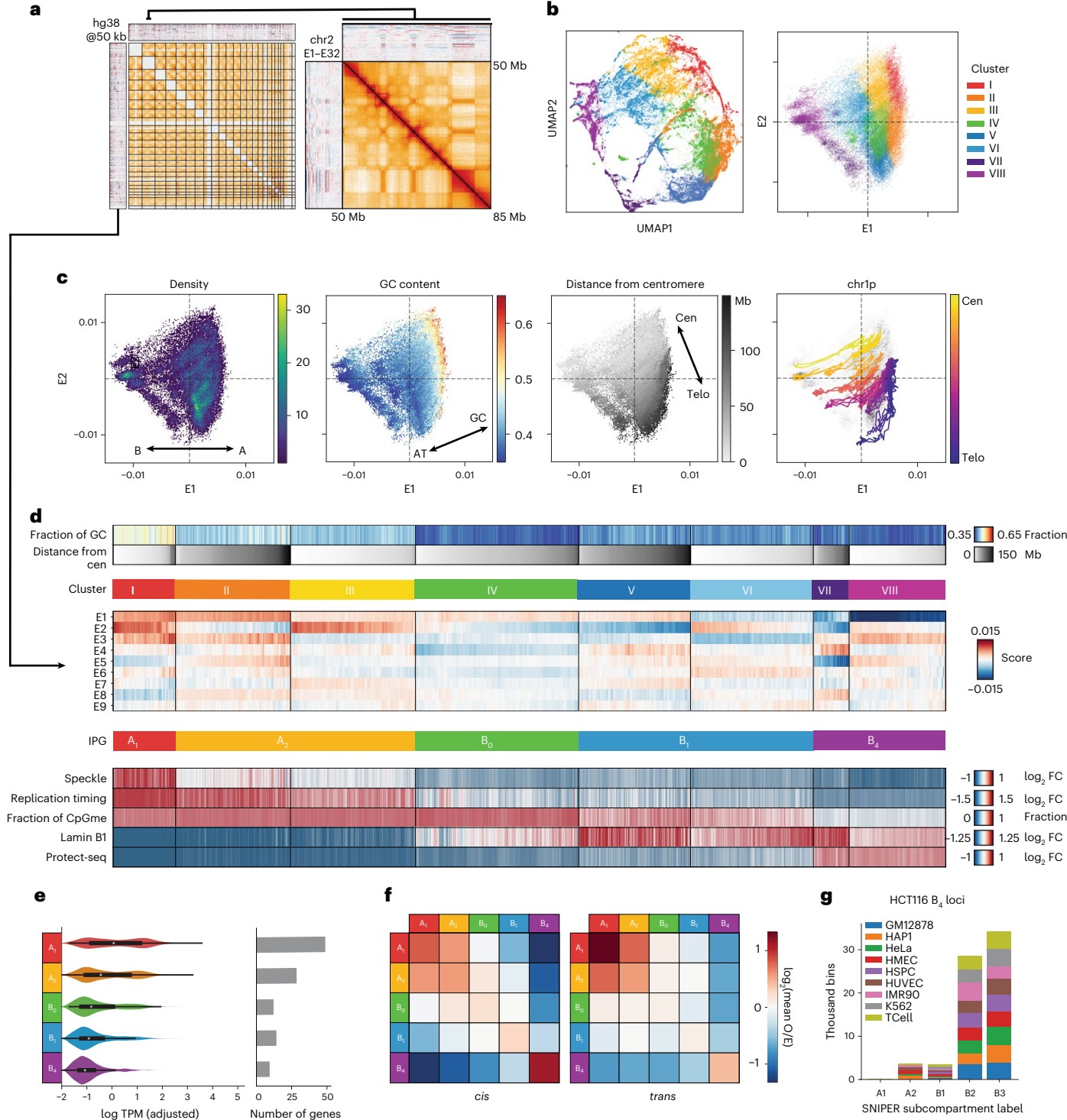

**Fig. 1 | Spectral decomposition of *trans* Hi-C data identifies distinct interaction profiles. a**, Left, map of *trans* Hi-C in HCT116 and heatmaps of leading *trans* eigenvectors; right, magnification of a 35-Mb intrachromosomal region of chr2. **b**, Scatter plots of all 50-kb genomic bins projected onto (left) a UMAP embedding of E1–E9 and (right) the E1–E2 subspace, colored by cluster identity (*k*-means, *k* = 8). **c**, E1 versus E2 scatter plots colored, from left to right, by point density, GC content and distance from the centromere. The fourth plot traces the trajectory of a single chromosome arm (chr1p) from centromere to telomere. **d**, Heatmaps of mean signal intensity of functional genomics features (rows) for each 50-kb genomic bin (column), grouped into Hi-C-derived clusters as in **b**. Top to bottom: GC content, distance from centromere, TSA-seq for SON, two-stage Repli-seq (Early/Late), fraction of methylated CpGs derived from WGBS, LaminB1 DamID-seq and Protect-seq. Clusters (I–VIII) are ordered by

ascending Protect-seq signal and within each cluster bins are sorted by distance from the centromere. Three pairs of clusters are combined for a total of five IPGs indicated in the lower row of colored bars. E1–E9 are displayed in between. **e**, Left, violin density plots of total RNA expression per IPG represented as log₁₀ transcripts per million (TPM) with internal miniature box plots. White circles mark the median, box bounds represent the interquartile range and whiskers extend by a factor of 1.5 for TPMs from IPGs A1 (*n* = 8,233 genes), A2 (*n* = 18,390), B0 (*n* = 5,369), B1 (*n* = 8,862), B4 (*n* = 3,200). Right, gene density in genes per megabase in each IPG. Same sample sizes as on the left. **f**, Pairwise mean observed/expected contact frequency between IPGs at 50 kb in *cis* (left) and *trans* (right). **g**, Distribution of SNIPER subcompartment label assignments[36] of HCT116 B₄ loci across various cell types. FC, fold change; O/E, observed/expected.

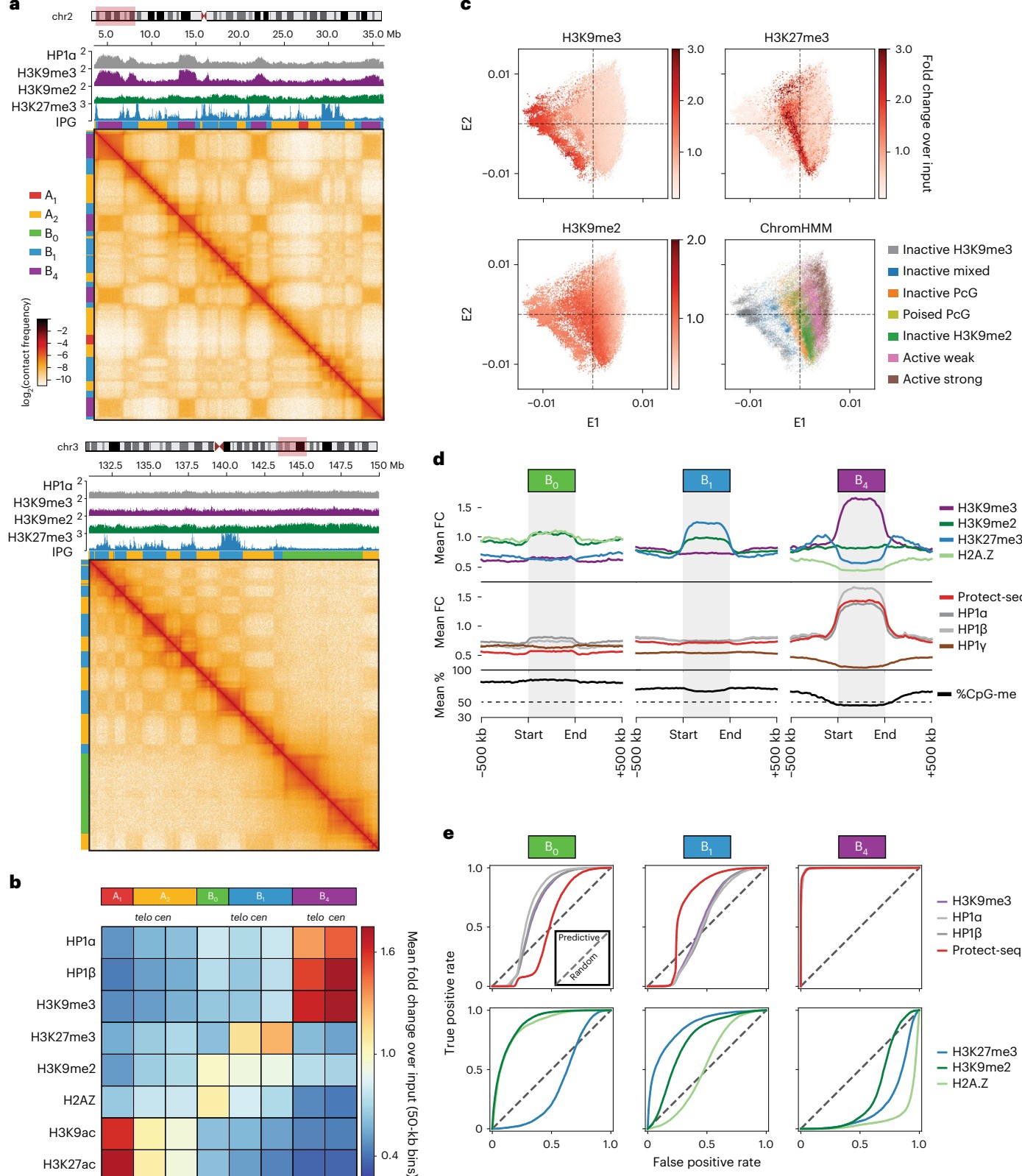

**Fig. 2 | 3D interaction profiles discern three types of silent chromatin in HCT116. a**, Two example regions illustrating the contrasting interaction profiles of $B_4$ domains (left, chr2:3.5–3.6 Mb) and $B_0$ domains (right, chr3:131–150 Mb) against $A_1$, $A_2$ and $B_1$ in *cis*. The IPG labels are displayed as colored bars on the top and left margins ($A_1$, red; $A_2$, yellow; $B_0$, green; $B_1$, blue; $B_4$, purple). Top, ChIP-seq tracks for HP1α, H3K9me2, H3K9me3 and H3K27me3. **b**, Heatmap of mean fold enrichment of ChIP-seq signal intensity for histone modifications, H2A.Z, and HP1α and HP1β proteins averaged over 50-kb bins in each interaction cluster ($k = 8$). **c**, Metaplots

of $B_0$, $B_1$, $B_4$ domains, rescaled to 25 bins and flanked by ±500 kb, displaying signal enrichment for ChIP-seq (H3K27me3, H3K9me2, H3K9me3, H2A.Z, HP1α/β/γ), Protect-seq and DNA methylation. **d**, E1–E2 scatter plots of 50-kb bins colored by ChIP-seq signal enrichment (H3K27me3, H3K9me2, H3K9me3) and ChromHMM state annotation. **e**, ROC curves assessing the prediction performance of individual 50-kb-aggregated functional tracks (ChIP-seq, Protect-seq) when treated as binary classifiers for $B_0$, $B_1$ or $B_4$ loci. The discrimination parameter in each case is a simple binarization threshold on the entire signal track.

despite a high degree of overlap between these loci in HCT116 and those labeled B2/B3 in GM12878, the corresponding chromatin states in the two cell types are different (see below). To indicate epigenetic similarity, we chose to assign the name $B_4$ to the IPG combining clusters VII and VIII since it appears to be most epigenetically similar to GM12878 subcompartment B4. Compared with $B_4$, loci in $B_1$ have more diverse subcompartment labels in different cell types, which is consistent with facultative heterochromatin (Extended Data Fig. 1b).

Interestingly, we identified an IPG (cluster IV) with no equivalent in GM12878, whose loci share hallmarks of inactive chromatin (Fig. 1d). Despite low GC content, it exhibits high CpG methylation frequencies and no Protect-seq enrichment (Fig. 1d). This IPG has a distinct 3D interaction profile, showing only modest preference for homotypic contacts (Fig. 1f), suggesting these do not form well-defined spatial subnuclear compartments. However, the regions of this IPG do form large continuous domains, present on many chromosomes (Extended Data Fig. 1c). When these loci are compared with subcompartment labels in other cell types they appear to be either weakly transcriptionally active (A2) or silent (B3) (Extended Data Fig. 1b), suggesting that this IPG could represent a 'poised heterochromatin' that transitions between active and inactive chromatin in different cell types. We termed this IPG $B_0$.

### Epigenomic data support three inactive IPGs in HCT116

To understand the chromatin composition of the IPGs, we examined histone modifications, histone variants and related factors (Fig. 2a). Consistent with $B_1$ being facultative heterochromatin, these loci are predominantly enriched for H3K27me3, with a mild enrichment in H3K9me2 (Fig. 2b,d). $B_0$ also displays a subtle enrichment in H3K9me2 and a Hidden Markov Model (HMM) (ChromHMM; Methods) showed that $B_0$ is almost entirely composed of H3K9me2 without H3K27me3 (Fig. 2b,d and Extended Data Fig. 3a,d). Loci in IPG $B_4$ are marked with H3K9me3, HP1α and HP1β, consistent with these loci being in a constitutive heterochromatic state (Fig. 2b,d and Extended Data Fig. 4b). Finally, when the E1–E2 projection of loci is colored by H3K27me3 or H3K9me3 an enrichment pattern spans the entire E2 axis, further validating the consolidation of centromere/telomere-proximal cluster pairs into functionally consistent IPGs (Fig. 2c and Extended Data Fig. 3c).

Curiously, in addition to H3K9me2, $B_0$ also has a mild enrichment for the histone variant H2A.Z (Fig. 2b,d). In humans, hypoacetylated H2A.Z has been reported to coexist with H3K9me2 in broad lamina-associated chromatin domains, suggesting that the $B_0$ IPG could correspond to a similar type of chromatin[45–47]. Moreover, $B_0$-like domains that display neutral interaction profiles in Hi-C, late replication timing and broad H2A.Z chromatin modifications can be observed in other cell types including primary cells (Extended Data Fig. 4a).

Our $A_1$ and $B_4$ IPG assignments (7.5% and 15.9% of the genome, respectively) exhibit the closest correspondence to known euchromatic and heterochromatic chromatin states, respectively. This can be observed using receiver operating characteristic (ROC) curves generated by using thresholded 50-kb binned signal tracks as binary classifiers for individual IPG assignments (Fig. 2e and Extended Data Fig. 3b). The $A_1$ label is predicted by the nuclear speckle marker SON with an area under the curve of 0.986, and the $B_4$ label is predicted by each of H3K9me3, HP1α, HP1β and Protect-seq with area under the curve > 0.992. These close correspondences, coupled with $A_1$ and $B_4$ being the most self-interacting IPGs, suggest that homotypic affinity between those marks or associated factors could be drivers of $A_1$ and $B_4$ compartmentalization. Other IPGs are less well predicted by any single chromatin modification, even though a particular histone modification may be globally enriched. The lack of contact enrichment between the different inactive IPGs ($B_0$, $B_1$, $B_4$) suggests that the homotypic interactions are specific to each type (for example, specific bridging proteins) rather than a generic form of interaction common to all inactive chromatin.

In summary, we discern three types of inactive chromatin by long-range contact frequencies in HCT116. Notably, none of these types appears to share an epigenetic similarity with the B2/B3 subcompartments described in GM12878 (Extended Data Fig. 2a,b). These results therefore hint at a greater diversity of inactive chromatin types, within and between cell types, than broadly attested.

### $B_4$'s chromatin state has varying cell-type abundance

Our data show that $B_4$ domains are enriched for H3K9me3, HP1α and HP1β and have strong homotypic interaction preferences. We next asked whether these properties are conserved in other cell lines. First, we examined enrichments of H3K9me2/3, HP1α/β/γ, H3K27me3 and H2A.Z and binned them into quantiles according to E1 value (Fig. 3a). K562 cells, similar to HCT116 cells, are enriched for H3K9me3, albeit more weakly (Fig. 3a and Extended Data Fig. 5a,b). In GM12878 cells we observed lower abundance of H3K9me3, and H3K9me3 was also found in active regions. Human embryonic stem cells (H1) have an even lower abundance of H3K9me3 (Fig. 3a), consistent with microscopy data suggesting H1 lacks punctate constitutive heterochromatin[48,49].

To understand whether the presence of H3K9me3, HP1α and HP1β was correlated with preferential homotypic interactions, we profiled *cis* contact frequency between pairs of loci ranked by their E1 eigenvector status and compared this with a ranking by H3K9me3 enrichment. Loci with similar E1 status tend to interact with each other, as expected (Fig. 3b), and loci that display high levels of H3K9me3 also show particularly high contact frequencies with each other (Fig. 3c and Extended Data Fig. 5c,d). This phenomenon is observed in all cell types even though GM12878 and H1 have a much lower abundance of H3K9me3 loci than HCT116. Loci in the highest H3K9me3 quantiles also show elevated HP1α in all cell types as well as HP1β where data were available (Fig. 3d). We conclude that the presence of H3K9me3 along with HP1α and HP1β is correlated with elevated homotypic contact frequency across cell types regardless of genomic abundance. Additionally, in GM12878 and K562 we also observe a coenrichment of HP1γ with H3K9me3, while HP1γ is anticorrelated with H3K9me3/HP1α in HCT116 (data for H1 were unavailable).

HCT116 cells have large ungapped H3K9me3 ($B_4$) domains up to several megabases in length (Fig. 3e and Extended Data Fig. 6a,b). Taking the largest domains ranked by size for each of the other cell types, we observe that K562 and fibroblasts (HFFc6, IMR90) also exhibit large domains. In GM12878 and H1 cells we observed shorter domains compared with HCT116 and K562. Yet even among the few domains in H1 cells displaying H3K9me3 and HP1α, we observe a tendency to self-interact (Extended Data Fig. 6c). It is noteworthy that, in contrast to *cis* contact frequency, *trans* contact frequency between H3K9me3-containing loci is not generally elevated across cell types (Extended Data Fig. 5c,d). These data argue that chromosomal territoriality and/or association with nuclear landmarks (for example, lamina) can limit the extent of interchromosomal contacts between H3K9me3 loci. Finally, the fact that loci with similar E1 values show preferred interactions with each other, across the full range of E1 values, indicates that other factors besides H3K9me3-HP1 can also mediate such interactions (Fig. 3b).

Taken together, these data suggest that the constitutive heterochromatin marks, H3K9me3 and HP1, define a homotypically interacting chromatin state, but that the prevalence and distribution of this chromatin state varies substantially across cell types. The exact combination of HP1 homologs and/or posttranslational modifications may govern the abundance and strength of the interactions[50].

### H3K9me3-HP1α/β chromatin is depleted for extrusion barriers

Besides compartmentalization, another major organizing mechanism in the nucleus is loop extrusion. The signature patterns of loop extrusion

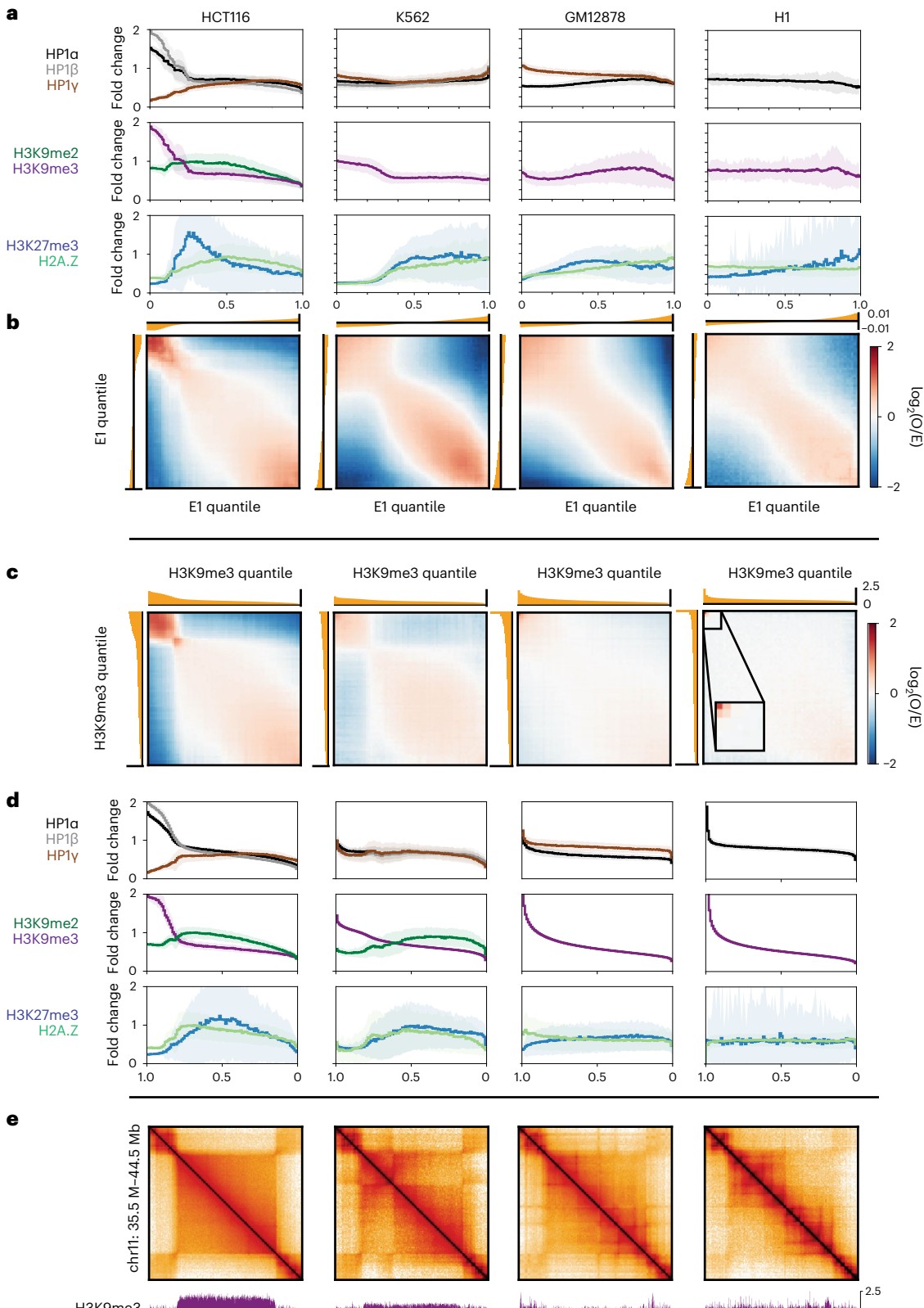

**Fig. 3 | Comparative analysis indicates a wide prevalence range of chromatin marked by H3K9me3, HP1α and HP1β and strong homotypic interaction preference.** Comparative analysis of genome organization and heterochromatic marks across HCT116, K562, GM12878 and H1-hESC. **a**, Histograms of ChIP–seq signal for repressive histone marks, HP1 proteins and H2A.Z grouped by eigenvector (E1) percentile and displayed in ascending order of E1 rank. Solid lines display the mean over the 50-kb bins within each percentile and include a standard deviation envelope. **b**, Bivariate summary maps of observed/expected contact frequency (also known as saddle plots) based on E1 percentiles and aligned with the univariate ChIP–seq histograms above in **a**. **c**, Bivariate summary maps similar to **b** but based on percentiles of H3K9me3 signal, displayed in descending order of H3K9me3 rank. **d**, Histograms of ChIP–seq signal similar to **a** but based on percentiles of H3K9me3 signal, aligned with the bivariate summary maps in **c**. **e**, Hi-C maps of a region containing a B₄ domain in HCT116 (chr11:34.5–44.5 Mb) and corresponding H3K9me3 signal below.

are fewer in number and less evident in $B_4$ domains in HCT116 cells. We therefore wanted to understand why these features are depleted and asked whether it is due to a lack of extrusion by cohesin, a lack of CTCF barriers or both.

First, we examined $B_4$ domains in cells with normal CTCF barriers but without cohesin-extruded loops (that is, cells depleted for Rad21 using an auxin-inducible degron approach)[10]. We looked at the decay of contact probability with genomic separation, $P(s)$, which is indicative of the underlying polymeric folding of the region[51]. We found that $P(s)$ was affected by depletion of cohesin in all IPGs, including $B_4$ domains, leading to the disappearance of the characteristic extrusion 'shoulder' in $P(s)$ (Fig. 4a)[52]. Moreover, we found that the shapes of the $P(s)$ derivatives suggest that $A_1$ and $A_2$ domains have more loops per kilobase than $B_4$ and that $B_4$ has a larger average loop size (Fig. 4a).

Second, despite $B_4$ domains appearing relatively featureless in Hi-C maps, we find that extrusion-related stripes and dots (which disappear upon cohesin depletion) originating outside a domain can sometimes propagate through it, appearing along the periphery of the square (Extended Data Fig. 7a). In the loop extrusion model, this would require the passage of extruded loops through the heterochromatic region, suggesting that heterochromatic regions are traversable by cohesin. To test whether the loop extrusion machinery can traverse $B_4$ domains, we turned to polymer simulations of loop extrusion in a heterochromatic domain surrounded by tandem CTCF clusters. Stripes extending along the periphery of the $B_4$ domains failed to appear when translocation of loop extrusion factors into such domains was blocked (Extended Data Fig. 7b).

Third, we find that the number and strength of CTCF peaks is depleted in $B_4$ domains compared with other IPG domains (Fig. 4b and Extended Data Fig. 7d,e). Concomitantly, we see fewer and weaker insulating loci in Hi-C at $B_4$ domains (Fig. 4c). Likewise, when we aggregate Hi-C data at CTCF-bound sites we find these sites form stripe-like features and local insulation (Fig. 4d). For CTCF-bound sites in $B_4$ domains these features are weak compared with those in other IPGs (Fig. 4a). In contrast, when we examine HCT116 $B_4$ regions in H1 human embryonic stem cells (H1-hESC), where H3K9me3-HP1α/β chromatin is lacking, we do not observe a similar reduction in number, occupancy or insulation of CTCF sites (Extended Data Fig. 7c–e). Altogether, our analysis argues that the low CTCF occupancy of $B_4$ domains in HCT116 is not intrinsic to the DNA sequence, but rather that $B_4$ domains in HCT116 are refractory to CTCF occupancy.

Finally, we also asked whether the depletions of extrusion features in H3K9me3-HP1α/β regions are conserved across cell types. While we find it generally to be the case, we do find a subset of heterochromatic domains that have both broad H3K9me3 enrichment and late replication timing, but also include extrusion-associated patterns in Hi-C (for example, normal human epidermal keratinocyte (NHEK) cells) (Fig. 4e). We predicted that this subset of domains should have occupied CTCF binding sites at regions of low H3K9me3 saturation. Indeed, the visible TAD boundary loci have lower H3K9me3, are enriched for H2A.Z and display narrow peaks for CTCF as well as marks such as H3K27ac and H3K27me3, suggesting that chromatin tends to be locally decompacted at these sites (Fig. 4e). These data are reminiscent of 'euchromatin islands' previously described as small regions of CTCF occupancy embedded within large heterochromatin domains[53]. The fact that dots and stripes can be detected in NHEK cells that cross domains enriched in H3K9me3 again shows that loop extrusion can traverse heterochromatin.

Altogether, these data suggest that the depletion of dots and stripes in $B_4$/H3K9me3-HP1α/β is the result of low CTCF occupancy, and not because of an absence of extrusion. The density of extrusion barriers differs across IPG domains, resulting in different average extruded loop sizes (Fig. 4f).

## DNMT perturbation selectively disrupts $B_4$ compartmentalization

Thus far we have defined the properties of H3K9me3-HP1α/β heterochromatin domains. We next wanted to understand how these features contribute to compartmentalization and chromatin state by disrupting these regions. To this end we chose to interrogate a double-knockout DNA-methylation-deficient HCT116 cell line (DNMT3b$^{-/-}$;DNMT1$^{-/-}$, hereafter referred to as DKO)[54] which has been shown to have defects in H3K9me3 (ref. 55) and HP1α/β deposition[37], in addition to perturbing DNA methylation in HCT116 cells by treatment with 5-Azacytidine for 48 h (5Aza) (Fig. 5a). In our hands, both conditions reduced DNA methylation compared with HCT116 cells as measured by LC–MS (Fig. 5b).

As we have previously shown, in DKO cells only a subset of domains are no longer detected by Protect-seq and no longer display HP1α and H3K9me3 binding, indicating that these domains are no longer in a closed heterochromatic state (Fig. 5c and Extended Data Fig. 8a)[37]. This shows that not all $B_4$ domains are equally sensitive to DNMT1/DNMT3b loss. Interestingly, in the 5Aza-treated cells we find that all H3K9me3-HP1α/β domains show mild but uniform depletion of both Protect-seq signal, and HP1α and H3K9me3 levels (Fig. 5c,d and Extended Data Fig. 8a).

To determine if loss of H3K9me3 affected self-affinity, we performed Hi-C on HCT116, DKO and 5Aza-treated cells. We ranked HCT116 $B_4$ domains by H3K9me3 loss in DKO and split them into those that lose H3K9me3-HP1α/β status in DKO cells (disrupted domains) and those that retain it (persistent domains) (Fig. 5e,f). Hi-C analysis shows striking local defects in $B_4$ compartmentalization (loss of checkering on the Hi-C map) and a global weakening of $B_4$ compartmentalization in 5Aza-treated cells (Fig. 5g,h and Extended Data Fig. 8d,f,g). Next, we aimed to investigate the interaction profile acquired by disrupted domains in DKO. Aggregate analysis of contact frequency shows that disrupted domains change to a more neutral interaction profile (Fig. 5h), reminiscent of the interaction profile of $B_0$ domains. We also examined the chromatin state at disrupted domains in DKO cells using available data for histone modifications and H2A.Z in DKO cells[37,55]. In contrast to persistent domains which maintain an H3K9me3-HP1α/β chromatin state, we find that disrupted domains transition to a chromatin state enriched for H3K9me2 and H2A.Z (Fig. 5i and Extended Data Fig. 8a–c,e), which is characteristic of $B_0$ domains.

## Late replication timing persists without H3K9me3-HP1α/β

Our data suggest that upon loss of DNA methylation, $B_4$ domains can lose H3K9me3, HP1 and self-affinity. Replication timing has been proposed to maintain the global epigenetic state in human cells[56]. In turn, histone deposition, HP1 proteins and DNMT1 are associated with chromatin restoration at the replication fork[57,58]. Therefore, we hypothesized that the loss of H3K9me3-HP1α/β heterochromatin in DKO cells would be accompanied by a change in the timing of DNA replication at disrupted domains. To address whether replication timing is altered by the disruption of heterochromatin, we performed two-stage Repli-seq in HCT116 and DKO cells. Surprisingly, we observe similar replication timing profiles between HCT116 and DKO cells (Fig. 6a and Extended Data Fig. 9a,b), consistent with recent findings using single-cell Repli-seq[59].

A fine-scale analysis of individual loci further shows that changes in replication timing and changes in the Hi-C E1 eigenvector are uncoupled (Fig. 6b,c). Both persistent and disrupted $B_4$ domains, which are late replicating in HCT116 cells, remain late replicating in DKO cells (Fig. 6b,e). Importantly, we do not see major early/late replication timing differences within disrupted $B_4$ regions (that is, that lose H3K9me3 and HP1 and cease to compartmentalize in DKO cells) or within regions where H3K9me3 and HP1 were gained in DKO (Fig. 6a,d). We further identified regions of differential replication timing and we find that those regions which transition to early replication timing in

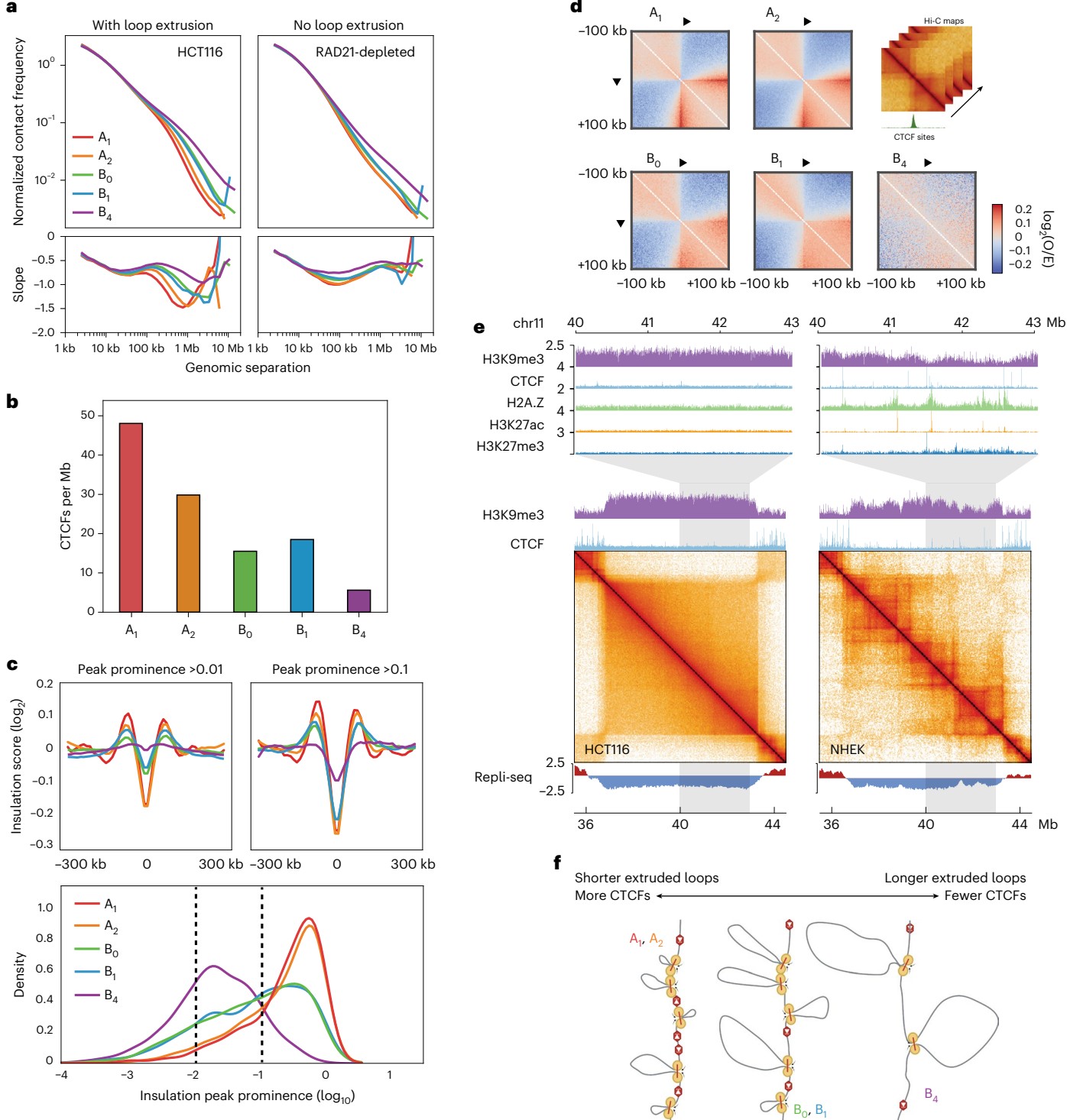

**Fig. 4 | All IPGs in HCT116 are permissive to loop extrusion but B$_4$ domains lack extrusion barriers. a**, Top, $P(s)$ curves, or interaction frequency as a function of genomic distance, for HCT116 and cohesin-depleted HCT116 RAD21-AID Hi-C, restricted to A$_1$ (red), A$_2$ (orange), B$_0$ (green), B$_1$ (blue), B$_4$ (violet), normalized to unity at $s = 10$ kb. Bottom, derivative of $P(s)$ indicating average sizes of extruded loops regardless of appearance of dots and stripes in Hi-C data. **b**, Mean number of CTCF peaks per megabase in each IPG. **c**, Top, average insulation score ($\log_2$) centered on 25-kb insulating loci (diamond size 100 kb) with peak prominence score (Methods) greater than or equal to 0.01 (left) and 0.1 (right) per IPG. Bottom, kernel density estimation plots of the insulation peak prominence ($\log_{10}$) distribution in each IPG. Dashed lines indicate cutoffs

for insulating loci used in panels above (>0.01 and >0.1 peak prominence). **d**, Average observed/expected Hi-C maps around CTCF binding sites within each IPG, centered at CTCF motifs oriented as indicated. Expected maps are calculated separately for each IPG. **e**, Contact frequency maps of a 9-Mb genomic region containing a B$_4$ domain in HCT116 (chr11:35.5–44.5 Mb) and the same region in NHEKs along with tracks for H3K9me3 and CTCF ChIP–seq above, and two-stage Repli-seq below. Top, magnifications of a 3-Mb subregion showing tracks for H3K9me3, CTCF, H2A.Z, H3K27ac and H3K27me3. **f**, Model of extrusion barrier (CTCF) sparsity determining the average extruded loop size as reflected in the $P(s)$ shoulder for each IPG, with B$_4$ domains having the fewest barriers and longest extruded loops.

DKO correlate with loss in H3K27me3, but not H3K9me3 (Extended Data Fig. 9c,d).

We find that replication timing in regions labeled $B_4$ in HCT116 is surprisingly insensitive to the presence or absence of H3K9me3-HP1α/β, despite the necessity of the H3K9me3-HP1α/β chromatin state for $B_4$ compartmentalization integrity. The fact that late replication is maintained in the absence of epigenetic and 3D signatures of heterochromatin implies that H3K9me3 and HP1 are not uniquely required to suppress the early onset of DNA replication and suggests alternative or compensatory mechanisms for maintaining late replication timing at disrupted domains. Motivated by this possibility, we investigated Hi-C and multistage (16-fraction) Repli-seq data from a recent study on the replication timing regulatory factor RIF1 (refs. [56],[60]). We found that while replication timing globally loses precision in the absence of RIF1, $B_4$ domains preserve very late replication timing (S12–S16 fractions) while $B_0$ domains shift from being moderately late in the wild type to predominantly early (Extended Data Fig. 9e,f). This suggests that the $B_0$-associated chromatin state depends on RIF1 for its late replication timing. Overall, these results support that disrupted $B_4$ domains in DKO cells transition to the late replicating silent chromatin state associated with the $B_0$ IPG.

### H3K9me3-HP1α/β heterochromatin suppresses CTCF binding sites

Our work thus far suggests that H3K9me3-HP1α/β domains cosegregate in the nucleus and permit loop extrusion, but are depleted in extrusion barriers. One striking observation in Hi-C data obtained with DKO and 5Aza-treated cells is the emergence of loop extrusion features (that is, extrusion barriers) in H3K9me3-HP1α/β domains, compared with HCT116 (Fig. 7a). Moreover, we observe an increase in insulating loci in all IPGs, suggesting that this is not limited to H3K9me3-HP1α/β domains but rather is a global phenotype (Extended Data Fig. 10b,c). Next, we aimed to understand the mechanism behind the gain of extrusion barriers.

It has been shown that CTCF binding to DNA can be blocked by DNA methylation[30],[31], and genome-wide loss of DNA methylation has been shown to increase CTCF occupancy at CpG-containing motifs (termed reactivated CTCF sites)[61]. Hence, we hypothesized that new loop extrusion features seen in DKO and 5Aza-treated cells are due to reactivated CTCF sites. To confirm that loss of DNA methylation reactivates cryptic CTCF sites, we performed chromatin immunoprecipitation (ChIP) followed by sequencing (ChIP–seq) in HCT116, DKO and 5Aza-treated cells. To identify high-confidence reactivated CTCF peaks, we chose overlapping reactivated CTCF peaks from DKO (this study), DKO (ref. [61]) and 5Aza (this study) not present in HCT116 ($n = 1,050$) (Extended Data Fig. 10a,d). Reactivated CTCF sites are present in all IPGs, consistent with our observation that the increase in extrusion barriers occurs globally (Fig. 7b). In accordance with the role of CTCF as a barrier to loop extrusion, we also see an enrichment of cohesin complex factors RAD21 and SMC3 at reactivated CTCF sites only in DKO and 5Aza-treated cells (Fig. 7b and Extended Data Fig. 10d,e). To further demonstrate that reactivated CTCF sites are functional as extrusion barriers, we generated aggregate heatmaps of Hi-C contact frequency centered at reactivated CTCF sites for each IPG (Extended Data Fig. 10b). As expected, we observe an increase in insulation in DKO and 5Aza compared with HCT116. In sum, these data support that loss of DNA methylation leads to the emergence of functional CTCF sites which can act as barriers to stall loop-extruding cohesin complexes.

To further investigate the genome-wide patterns of CTCF reactivation, we profiled DNA methylation, chromatin inaccessibility and histone modifications in relation to IPGs. To our surprise, reactivated CTCF motifs within $B_4$ regions lack CpG methylation in normal untreated HCT116 cells, in contrast to motifs in all other IPGs (Fig. 7c and Extended Data Fig. 10f). These data suggest that DNA methylation could regulate CTCF via two mechanisms: direct and indirect. The direct mechanism relies on canonical CpG methylation within the core motif[30],[31],[33],[61–63], while the indirect mode of regulation within $B_4$ is likely independent of motif methylation. Consistent with this observation, CTCF motifs within $B_4$ contain lower CpG dinucleotide frequencies than the consensus core motif (Extended Data Fig. 10g). We speculate that this mechanism acts through nucleosome occlusion, which is consistent with the strong H3K9me3, Protect-seq and HP1α/HP1β signal directly over the CTCF motif (Fig. 7d and Extended Data Fig. 10e). In agreement with our results, increased CTCF occupancy was observed in *Setdb1*-deficient mouse neurons[64], and a similar 5-methylcytosine (5mC)/nucleosome occlusion model has been proposed to regulate CTCF binding in mouse embryonic stem cells[65],[66].

## Discussion

Our study demonstrates a remarkable cell-type-related diversity in inactive chromatin and its relationship to 3D genome organization. In HCT116, each of the three inactive IPGs exhibits a distinct chromatin state, Protect-seq signal and DNA methylation status, and displays differences in homotypic affinity and the regulation of loop extrusion barriers (Table 1). The existence of cell-type-specific chromatin and contact frequency profiles highlights the need for de novo assessment of any given cell type. Our approach identified the $B_0$ IPG in HCT116 cells which is not observed in GM12878 cells, forming large domains that do not display strong homotypic interactions. Yet another inactive chromatin state appears to underlie the B2/B3 subcompartments in GM12878 and remains poorly characterized. Notably, the features originally reported as enriched in B2 and B3 came from dissimilar cell types: HeLa[67], HT1080 fibrosarcoma[68] and skin fibroblasts[69]. Elucidating the molecular intermediates determining the behavior of known and novel IPGs will require a combination of unsupervised techniques and deep chromatin profiling[70–72].

Our results reveal striking connections between DNA methylation, H3K9me3 and HP1 deposition, and 3D chromosome organization at the level of chromosome compartmentalization and loop extrusion.

---

**Fig. 5 | Inhibition or knockout of DNA methyltransferases disrupts H3K9me3-HP1α/β heterochromatin and compartmentalization. a**, Schematic of the DNA methylation perturbation system used in this study. **b**, LC–MS quantification of 5-methylcytosine/total cytosine for HCT116 (left, $n = 5$ biological replicates), HCT116 cells treated with 5Aza (48 h) (middle, $n = 10$) and DNMT1/DNMT3b knockout (DKO) (right, $n = 2$) cells. Data are presented as mean values with ±s.d. error bars. **c**, Stacked heatmaps of H3K9me3 ChIP–seq signal in HCT116 (left), 5Aza 48 h (middle) and DKO (right) centered at uniformly rescaled $B_4$ domains, sorted vertically by the intradomain H3K9me3 ratio between DKO and HCT116, and partitioned into two categories: persistent domains (top) and disrupted domains (bottom) in DKO. **d**, Scatter plots of 50-kb bins along E1 versus E2 (HCT116 eigenvectors), colored by Protect-seq signal for HCT116 (left), 5Aza 48 h (middle) and DKO (right). **e**, Box plots quantifying the distribution of $\log_2$ ratios of mean domain signal between HCT116 and 5Aza in persistent (left, $n = 185$) and disrupted (right, $n = 116$) $B_4$ domains. Signals shown are Protect-seq, HP1a, H3K9me3 and H3K9me2. Box extents give the interquartile range with whiskers extending by a factor of 1.5 and the notch representing the confidence interval around the median. Points represent outliers. **f**, Same as **e** but between HCT116 and DKO. **g**, Contact frequency maps of a 40-Mb genomic region (chr9:0–40 Mb) in HCT116 (left), 5Aza 48 h (middle) and DKO (right) containing representative examples of persistent and disrupted domains. Below, ChIP–seq tracks for H3K27me3, H3K9me2, H3K9me3, HP1α and H2A.Z. **h**, Heatmap displaying the pairwise mean observed/expected contact frequency between active, H3K27me3 and H3K9me3 domains split into either disrupted or persistent labels in DKO based on ChromHMM states learned at 50 kb. **i**, Sankey plot of disrupted domains illustrating the chromatin transition from H3K9me3-HP1α/β in HCT116 cells to H3K9me2 and/or other repressive states based on ChromHMM in DKO cells.

---

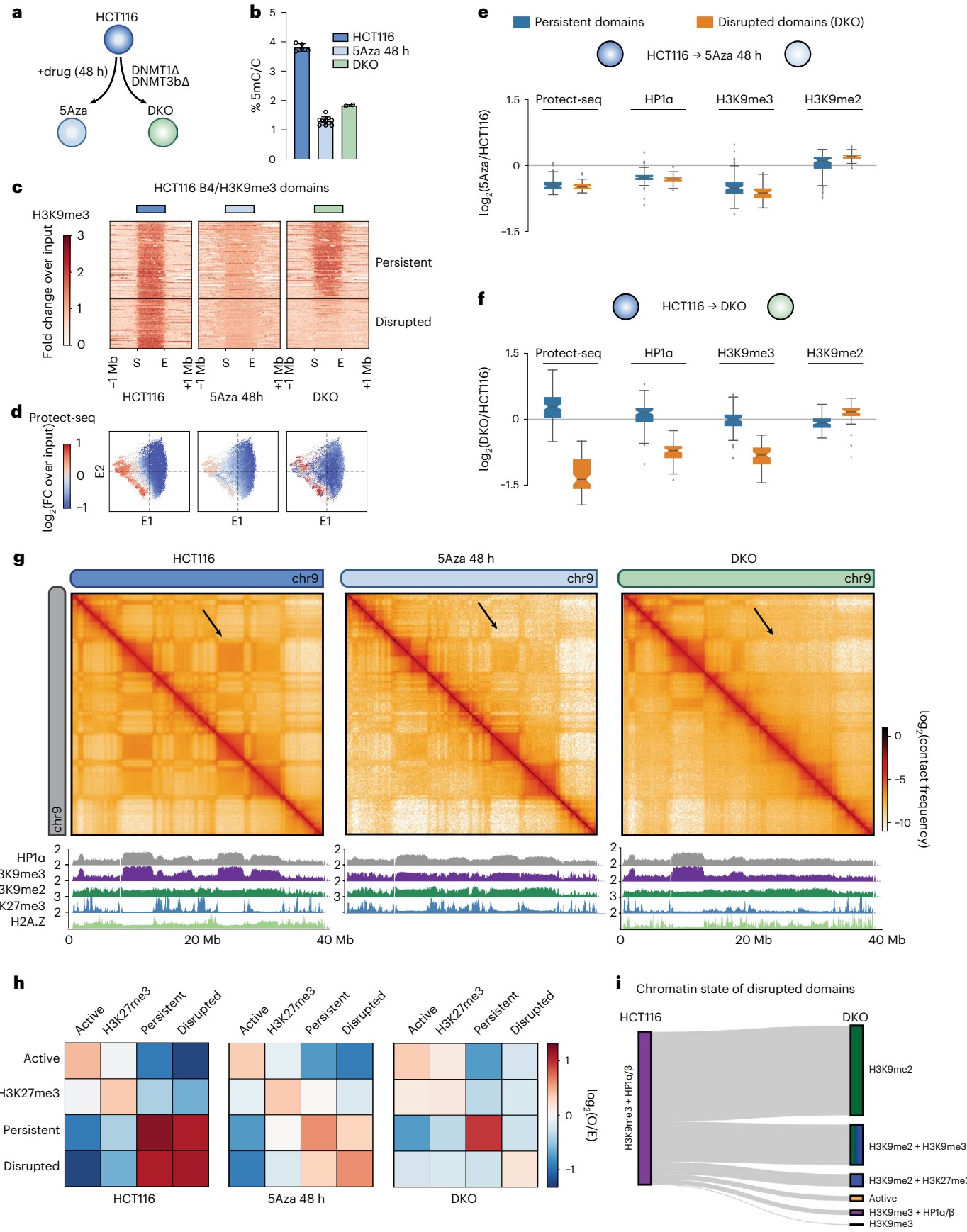

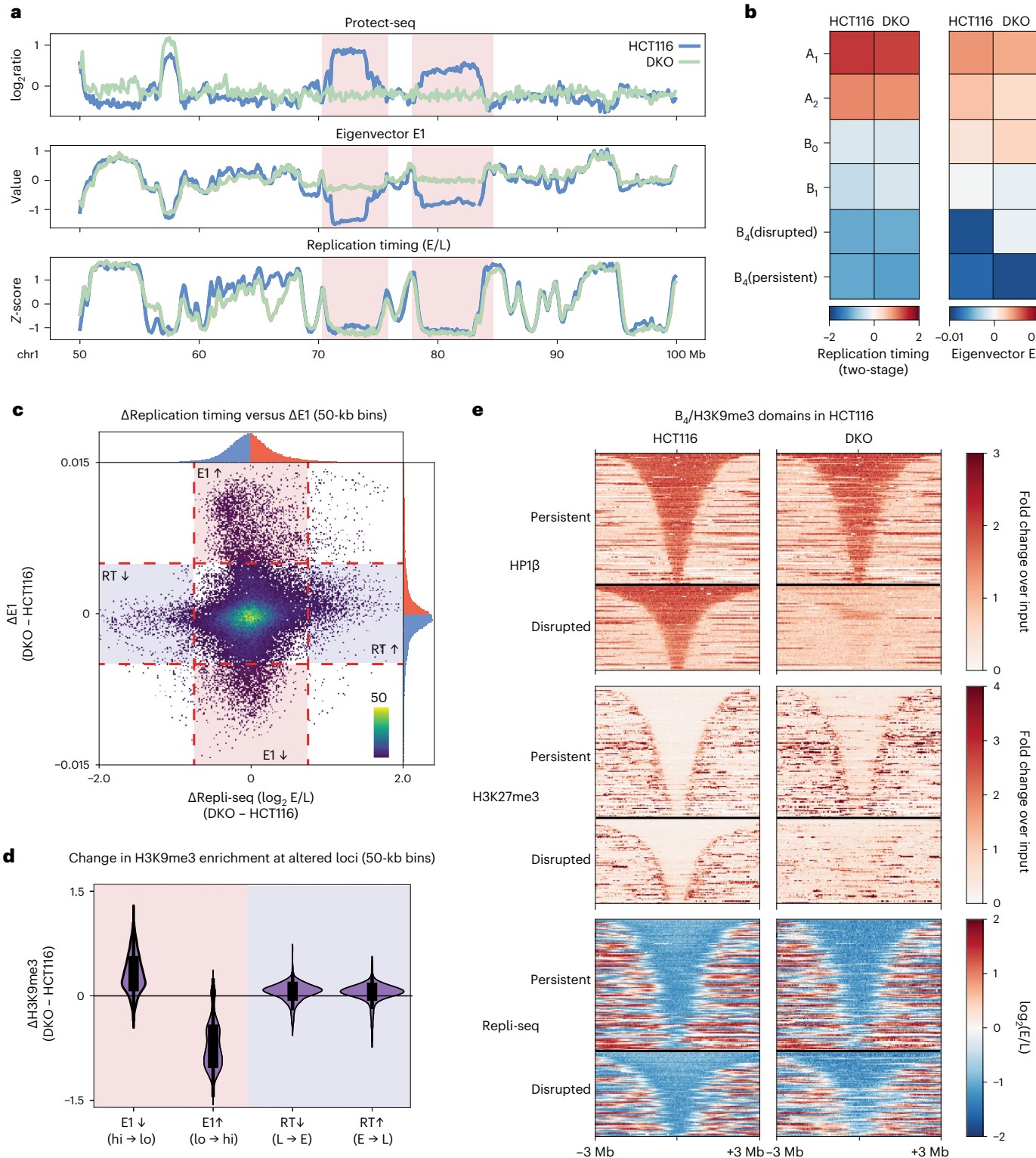

**Fig. 6 | Loss or gain of H3K9me3-HP1α/β is not correlated with replication timing alterations. a**, Example region (chr1:50–100 Mb) containing two disrupted domains in HCT116 cells (light blue) and DKO cells (light green) illustrating dramatic changes in compartmentalization without changes in replication timing. Top, Protect-seq signal track (log$_2$ signal/input). Middle, eigenvector track (E1). Bottom, two-stage Repli-seq shown as Z-score of log$_2$(Early/Late). **b**, Heatmaps of mean signal of Repli-seq (left) and E1 (right) over 50-kb bins per IPG in HCT116 and DKO. **c**, Scatter plot of change in E1 score versus change in Repli-seq signal for 50-kb bins (DKO − HCT116). Tail areas of uncorrelated variation of E1 and replication timing are gated and

shaded. **d**, Violin plots quantifying changes in H3K9me3 (DKO − HCT116) over groups of altered 50-kb bins depicted in **c**: decreased E1 score in DKO ($n = 2{,}167$ bins), increased E1 score in DKO ($n = 3{,}246$), decreased Early/Late signal in DKO (delayed replication timing, $n = 932$), increased Early/Late signal in DKO (hastened replication timing, $n = 1{,}501$). Box extents give the interquartile range with whiskers extending by a factor of 1.5. **e**, Stacked signal heatmaps of HP1β ChIP–seq, H3K27me3 ChIP–seq and Repli-seq in HCT116 (left) and DKO (right) centered at persistent (top) and disrupted (bottom) B$_4$ domains sorted vertically by size and flanked by ±3 Mb. E/L, Early/Late; RT, replication timing.

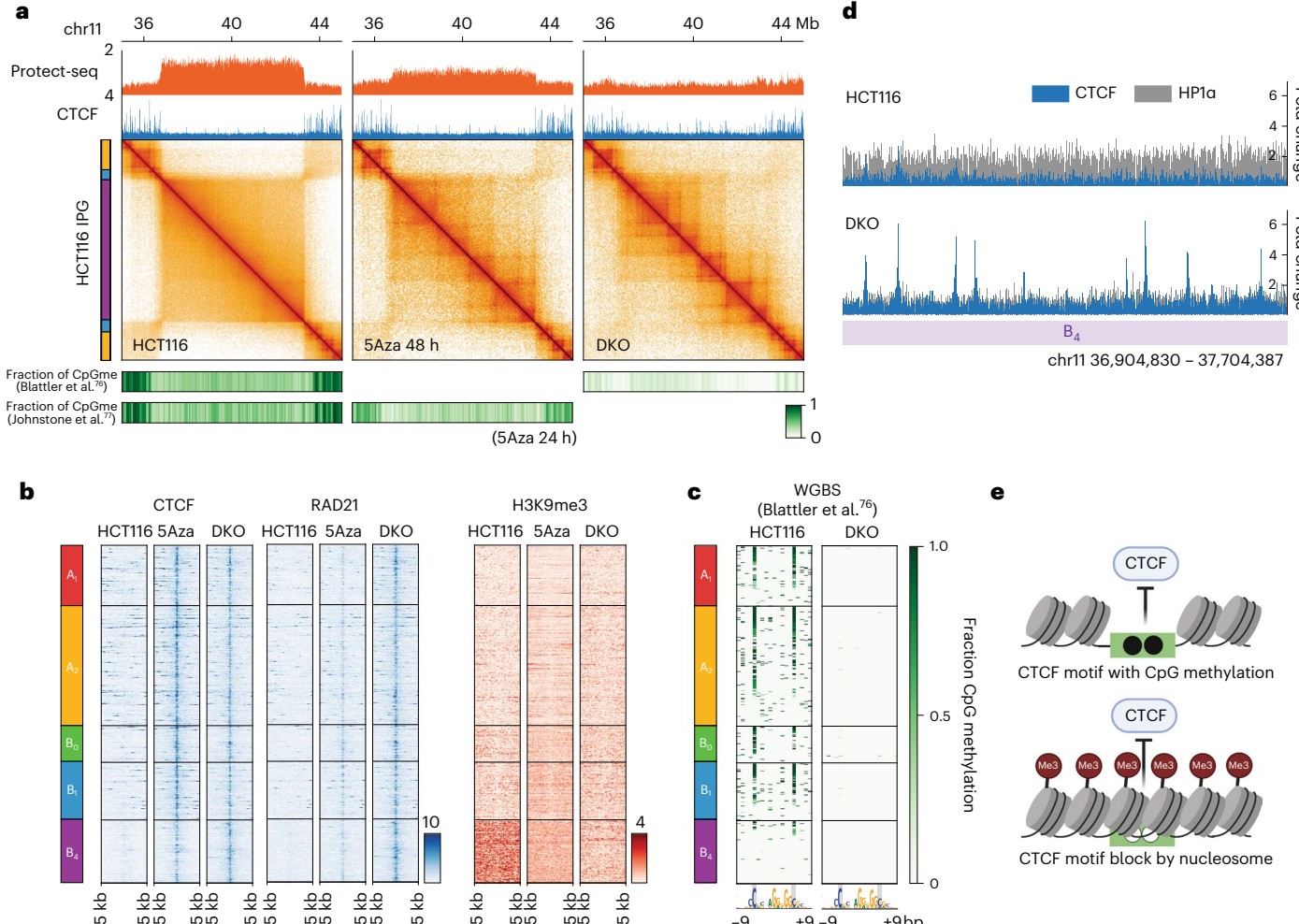

**Fig. 7 | Two modes of CTCF binding suppression depend on DNA methylation.** **a**, Contact frequency maps of a 9-Mb genomic region (chr11:35.5–44.5 Mb) in HCT116 (left), 5Aza 48 h (middle) and DKO (right) containing a representative example of reactivated CTCF sites. Top, Protect-seq and ChIP–seq track for CTCF. Bottom, heatmaps of CpG methylation frequencies in 50-kb bins from ref. 76 and ref. 77. **b**, Stacked heatmaps of reactivated CTCF sites for HCT116, 5Aza and DKO cells centered on the CTCF motif displaying ChIP–seq signal for CTCF (left), Rad21 (middle) and H3K9me3 (right) flanked by ±5 kb and segregated by IPG. **c**, Stacked heatmaps around reactivated CTCF site core motifs (19 bp) for

HCT116, 5Aza and DKO cells displaying fraction of CpG methylation using whole genome bisulfite sequencing data[76]. **d**, Example of reactivated CTCF sites within a $B_4$ domain (chr11:36.9–37.7 Mb). Top, HCT116 ChIP–seq signal for CTCF (blue) and HP1α (gray) overlayed. Bottom, DKO ChIP–seq signal for CTCF (blue) and HP1α (gray) overlayed. **e**, Model of two modes of CTCF regulation. Top, the direct mode involves CpG methylation within the core binding motif. Bottom, the indirect mode involves steric occlusion of the CTCF binding site by nucleosomes and/or other heterochromatic factors.

## Table 1 | Properties of inactive IPGs in HCT116

| | $B_0$ | $B_1$ (facultative) | $B_4$ (constitutive) |
|---|---|---|---|
| Histone modifications | H3K9me2 | H3K27me3 | H3K9me3 |
| Factors/histone variants | H2A.Z | PRC2, EZH2 | HP1α, HP1β |
| Replication timing (RT) | late | late | very late |
| RIF1-sensitive RT | + | – | – |
| CpG methylation | + | – | –– |
| 3D self-affinity | – | + | ++ |
| Protect-seq | – | + | ++ |
| Cohesin permissive | + | + | + |
| CTCF permissive | + | + | – |
| LaminB1/B2 | + | + | + |

Symbol legend: ––, very low; –, low; +, high; ++, very high.

We show that the heterochromatic state is integral to its nuclear compartmentalization. When DNA methylation is lost, H3K9me3-HP1/$B_4$ domains transition to the $B_0$-associated inactive chromatin state that lacks self-affinity yet maintains late replication timing. Whether this transition occurs via active chromatin remodeling or passive loss of heterochromatin remains unclear. It may be that the chromatin states underlying the $B_0$ (poised) and $B_4$ (constitutive) IPGs are generally transposable. Since disrupted $B_4$ domains acquire H2A.Z, it is possible that the presence of this histone variant could function to interfere with heterochromatin deposition and spreading, as has been shown in budding yeast[73]. Moreover, since $B_0$ selectively marks RIF1-sensitive late replicating chromatin in HCT116, it is possible that the $B_0$-associated chromatin state itself depends on RIF1.

Loop extrusion and compartmentalization shape different aspects of genome organization. While the forces driving compartmentalization are believed to be global and intimately linked to the state of chromatin, the degree to which loop-extruding cohesins are

influenced by the epigenome is not well understood. As loop extrusion has been shown to reduce the strength of compartmentalization and interfere with the segregation of short compartmental domains[10,12,14,15], our results represent a complementary phenomenon: strongly compartmentalizing heterochromatin suppressing the imposition of extrusion barriers (CTCF-bound sites) while remaining permissive to extrusion. These results highlight the two-way interplay between compartmentalization and extrusion.

The classic definition of heterochromatin originated from staining mitotic chromosomes[74] and later came to be associated with histone modifications[75]. We now have a more nuanced understanding of the molecular details, including several types of repressive histone modifications and associated proteins and their genomic distributions across cell types. Our work begins to unravel the diversity and plasticity in silent chromatin and its influence on genome compartmentalization, nuclear architecture and other chromosome-organizing processes.

## Online content

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

## Methods

### Cell culture

HCT116 and DKO cells were cultured in McCoy5A medium. DKO cells were grown in the presence of G418, geneticin. All media were supplemented with 10% FBS at 37 °C and 5% CO$_2$. For drug treatment, HCT116 cells were treated with 5 µM 5Aza for 48 h, then washed with 1 × PBS before collection.

### Crosslinking and nuclei preparation

Cells were grown to ~75% confluency, collected with trypsin, washed in 1× PBS and frozen/stored at −80 °C. Thawed cells were fixed in 1% formaldehyde and quenched in 0.125 M glycine, then washed twice in 1 × PBS. Fixed cells were then resuspended in 500 µl of lysis buffer (50 mM Tris-HCl pH 8.0, 10 mM NaCl, 0.2% NP40, 1 × protease inhibitor cocktail (PIC)) for 30 min on ice with periodic resuspension. Lysed cells were spun at 3,500 r.p.m. for 3 min and resuspended in 300 µl of 1 × NEB buffer 2, spun and resuspended in 198 µl of 1 × NEB buffer 2. Next, 2 µl of 10% SDS was added and incubated at 65 °C for 10 min. Afterwards, 400 µl of 1 × NEB buffer 2 and 60 µl of 10% Triton X-100 were added to quench the SDS. Samples were incubated at 37 °C for 15 min. Nuclei were spun at 3,500 r.p.m. for 3 min and resuspended in 300 µl of 1 × NEB buffer 2, and the wash step repeated.

### Protect-seq protocol

The Protect-seq protocol was performed as described in ref. 37. Pelleted nuclei were resuspended in 183 µl of DNaseI Buffer, then 2 µl of 100 mM Ca$^{2+}$ (1 mM final), 5 µl of DNaseI (10 U), 5 µl of MNase (10,000 U) and 5 µl of RNase A (20 mg ml$^{−1}$) were added (200-µl final volume). Cells plus the enzyme cocktail were incubated at room temperature (also works at 37 °C) for 30 min. Digested cells were spun at 3,500 r.p.m. for 3 min and resuspended in 400 µl of 1 × NEB buffer 2, then rotated at room temperature for 15 min. Digested/wash no. 1 cells were spun at 5,000 r.p.m. for 3 min and resuspended in the same 200 µl of cocktail mix and incubated again at room temperature (or 37 °C) for 30 min. Digested cells no. 2 were spun at 10,000 r.p.m. for 3 min and resuspended in 400 µl of 1 × NEB buffer 2, then rotated at room temperature for 15 min (save aliquot for microscopy). Then we spun digested cells no. 2 at 10,000 r.p.m. for 3 min and resuspended in 200 µl of 1 × NEB buffer 2, 20 µl of Proteinase K (SDS optional). They were digested overnight at 65 °C then purified using phenol/chloroform and ethanol precipitation (compatible with silica-bead purification).

### Illumina library preparation

DNA was quantified with Qubit (high-sensitivity) and sonicated using Covaris 50-µl, 300-bp protocol. Illumina libraries were prepared using the NEB Ultra II DNA library kit using the manufacturer's protocol. We used 4–5 PCR cycles to amplify next-generation sequencing (NGS) libraries and index samples.

### In situ Hi-C

The Hi-C protocol was performed similarly to ref. 4. In brief, fixed nuclei were isolated and digested with MboI (NEB no. R0147M), 5' overhangs were filled-in with a biotinylated nucleotide, blunt-ends were ligated, followed by reverse crosslinking overnight. The purified DNA (2 µg) was sonicated using Covaris 50-µl, 400-bp protocol. The sonicated DNA was brought to a volume of 400 µl in binding buffer (5 mM Tris-HCl pH 7.5; 0.5 mM EDTA; 1 M NaCl) and mixed with 20 µl of streptavidin magnetic beads (NEB no. S1421) and rotated for 1 h at room temperature. The bead-bound DNA was washed twice with 400 µl of low-TE (10 mM Tris-HCl (pH 8.0) + 0.1 mM EDTA) and resuspended in 50 µl of low-TE. Next-generation sequencing (NGS) libraries were prepared using NEB DNA Ultra II kit (NEB no. E7645). End prep: mixed 50 µl of sample with 7 µl of End prep buffer and 3 µl of End prep enzyme, incubated for 30 min at room temperature then 30 min at 65 °C, washed twice with 400 µl of low-TE and resuspended in 60 µl of low-TE. Adapter ligation:

2.5 µl of adapter and 30 µl of ligation mix were incubated at room temperature for 1–3 h, washed twice with low-TE and resuspended in 90 µl of low-TE; following ligation, 3 µl of USER was added for 30 min at 37 °C, washed twice with 400 µl and resuspended in 15 µl. PCR: added 5 µl of universal F and index R primer, 25 µl of Q5 mix, 15 µl of sample for 5 PCR cycles. Libraries were purified with SPRI beads (0.9×) and quantified on a bioanalyzer and with NEB Illumina Quant kit (NEB no. E7630). Hi-C libraries were sequenced on a NextSeq500, either 150-bp or 75-bp paired-end reads.

### ChIP experiments

SimpleChIP Plus Enzymatic Chromatin IP Kit (Magnetic Beads) no. 9005 from Cell Signaling Technologies was used for all ChIP–seq experiments, using the manufacturer's recommended protocol. We used 4 million cells per immunoprecipitation. Digested chromatin was pooled into a single tube for brief sonication to lyse nuclei. Supernatant was then split evenly between immunoprecipitations (minus 2% input). Antibodies and chromatin were incubated overnight at 4 °C, rotating. DNA was purified using spin columns and prepared using NEB Ultra II DNA Library Kit.

### Repli-seq

Repli-seq was performed and analyzed as described in ref. 78. In brief, cells were pulsed with 100 µM BrdU for 2 h, trypsinized, ethanol fixed, stained with propidium iodide and FACS sorted (SONY SH-800) based on DNA content (early S versus late S). Genomic DNA was purified using Zymo DNA Clean & Concentrator and sonicated on a Covaris (S2) using the 300-bp, 50-µl protocol. Libraries were made with Ultra II DNA kits from NEB and sequenced on an Illumina miSeq and/or nextSeq.

### Computational analysis

**Hi-C data processing.** Hi-C libraries were trimmed with the fastp package[79] to remove low-quality reads and sequencing adapters. Hi-C datasets were processed using the distiller pipeline (https://github.com/open2c/distiller-nf) written for nextflow[80]. Briefly, we mapped Hi-C sequencing reads to the human reference assembly hg38 using bwa mem (ref. 81) with flags -SP. Alignments were parsed, filtered for duplicates and pairs were classified using the pairtools package (https://github.com/open2c/pairtools). Hi-C pairs were aggregated into contact matrices in the cooler format using the cooler package at multiple resolutions[82]. All contact matrices were normalized using the iterative correction procedure[35] after bin-level filtering.

**ChIP–seq and Protect-seq data processing.** All ChIP–seq data, including data from ref. 55 and ref. 61 but excluding those obtained from the ENCODE portal, were processed following the steps of the ENCODE ChIP–seq pipeline (https://github.com/ENCODE-DCC/chip-seq-pipeline2) with slight modifications using a simplified custom snakemake workflow. Briefly, reads were mapped to hg38 using bwa mem (ref. 81). Alignment files (BAM format) were filtered for quality and duplicates using the samtools and Picard packages[83]. Cross-correlation analysis and fragment length estimation for single-ended datasets were performed using the phantompeakqualtools package[84]. Signal track (target over input) generation was performed using MACS2 (ref. 85). For CTCF, a motif instance was assigned to each ChIP–seq peak by scanning the core motif PWM (JASPAR MA0139.1) using gimmemotifs (ref. 86). Protect-seq data were mapped following the same procedure to produce signal tracks (treatment over input).

**Repli-seq data processing.** Two-stage Repli-seq reads were processed following the protocol described in ref. 78. Replicates were merged to produce signal tracks of log$_2$ count-normalized ratios of early divided by late fractions binned at 50-kb resolution. Tracks were then normalized by z-score transformation.

**Spectral analysis.** To characterize long-range interaction profiles, 50-kb resolution Hi-C maps were dimensionally reduced by applying global eigendecomposition on *trans* contact frequencies. First, we manually identified and excluded three large translocated segments in HCT116 based on published karyotype analysis[87] narrowed down by visual inspection of Hi-C data in HiGlass[88]. Structural variations in DKO, on the other hand, were too widespread to systematically exclude so DKO clustering results were omitted from this study. Next, to mask the influence of *cis* data, we followed the same procedure described in ref. 35, where *cis* pixels in the contact matrix are replaced with randomly sampled pixels from the same row or column. The resulting matrix was then re-balanced and scaled such that rows and columns summed to 1. Finally, the leading eigenvalues and associated eigenvectors of this matrix were then calculated using the eigsh routine from numpy, in descending order of eigenvalue modulus (that is, not respecting algebraic sign).

We describe our clustering method in more detail in the Supplementary Note. In summary, $m$ leading eigenvectors were rescaled and concatenated as columns, and $k$-means clustering was applied to the rows using scikit-learn. We produced cluster assignments for a range of $k$ for Hi-C maps of GM12878 (ref. 4), and both unsynchronized untreated and unsynchronized 6-h Auxin-treated Rad21-AID HCT116 (ref. 10), calculated silhouette scores (Extended Data Fig. 1) and visually compared cluster profiles with a large number of independent genomic tracks. The final number of clusters was chosen based on a balance of clustering metrics and interpretability.

For visualization of the approximate manifold structure, further dimensionality reduction on the $m$ leading eigenvectors was performed using UMAP[89]. Additionally, direct visual inspection of the unreduced eigenvector subspaces (pairwise) and related genomic and functional data proved to be indispensable for interpretability of clusters (see below).

**Rasterized scatter plots.** The new matplotlib (ref. 90) extension for the data graphics pipeline datashader (ref. 91) (dsshow function) (https://datashader.org) was used to generate scatter plot visualizations of points representing 50-kb genomic bins. The datashader pipeline is used to prevent overplotting dense point clouds by aggregating points onto a regular two-dimensional grid and either (1) color-mapping the resulting raster to associated quantitative values (for example, point count, mean value) or (2) displaying associated color-coded categorical values (cluster labels, chromosome and so on) via image compositing.

**ChromHMM state assignment.** We ran ChromHMM (ref. 92) to create epigenomic segmentations for HCT116 and DKO using bam files for ChIP–seq of broad marks/factors HP1a, HP1b, H3K9me3 and H3K27me3. For HCT116, we also included data for SON tyramide signal amplification sequencing (TSA-seq)[93]. Tracks were binarized at 50 kb using BinarizeBam and were modified to ignore bins filtered in Hi-C data. Models were trained using 50-kb bins (LearnModel -b 50000) for a range of state numbers. A seven-state model was chosen for HCT116. For DKO, a six-state model was able to qualitatively capture the same repressive states based on emission parameters (with only a single active state, since TSA-seq was not available to discriminate between two active states).

**Chromatin state analysis.** A gene quantification table for HCT116 was obtained from ENCODE and cross-referenced to GENCODE v29 basic gene annotations for hg38. Records were intersected against IPG labels using bioframe (ref. 94) and grouped. Adjusted transcripts per million values were log-transformed and violin and box plots were generated using seaborn (ref. 95).

HCT116 and DKO Whole Genome Bisulfite sequencing data (hg19) from ref. 76 were lifted over to hg38 using Crossmap (ref. 96). DNA methylation tracks for HCT116 and 5Aza-treated cells (24 h) generated using Hybrid Selection Bisulfite Sequencing (hg19) from ref. 77 were also lifted over to hg38 using Crossmap. All data were filtered for CpG context to exclude liftover base changes. A custom script was used to aggregate records into 50-kb bins and calculate the cumulative methylation fraction from CpGs divided by total number of CpGs per bin.

Functional profiles for spectral clusters (as in Fig. 1d, and averages in Fig. 2b) were derived from categorical or mean-aggregated quantitative signal tracks (distance from centromere, LaminB1 DNA adenine methyltransferase identification and sequencing (DamID-seq), SON TSA-seq, Protect-seq, Repli-seq, whole genome bisulfite sequencing (WGBS), ChIP–seq) at 50-kb resolution to match the resolution of IPG analysis.

IPG domain metaplots and stacked signal heatmaps were generated from BigWig files using the pybbi package (https://github.com/nvictus/pybbi). Unscaled stacked heatmaps were defined using the domain midpoints as a reference point flanked by a fixed genomic distance left and right, while rescaled stacked heatmaps were generated by independently partitioning the intradomain signal and flanking regions into a fixed number of bins. Metaplots were generated by averaging rescaled heatmaps vertically.

Sankey plots were generated by using ChromHMM segmentation maps from DKO cells. Chromatin states were intersected against disrupted domains using bioframe. Next, total base pairs overlapped for each chromatin state were counted. Sankey plots were generated using plotly.

**ROC curves.** To assess the correspondence of individual signal tracks to IPG assignments derived from Hi-C data, we treated each mean-aggregated 50-kb resolution track as a binary classifier to predict a given IPG label (one of $A_1$, $A_2$, $B_0$, $B_1$, $B_4$) by applying a simple value-based discrimination threshold on the signal track. ROC curves and area under ROC for these classifiers were calculated using scikit-learn. Curves that dip below the diagonal indicate thresholds with predictive power for the complement of the target label (for example, 'not $A_1$').

**Quantile-based ChIP–seq histograms and Hi-C summary maps.** The 50-kb-resolution ChIP–seq tracks were grouped into percentiles of either E1 signal or H3K9me3 signal to generate histograms and standard deviation envelopes.

Expected contact frequency versus distance profiles were generated using cooltools (ref. 97) (https://github.com/open2c/cooltools) and bivariate summary maps of observed/expected contact frequency (also known as saddle plots) using percentiles of either E1 or H3K9me3 signal as bins were also generated using cooltools.

**H3K9me3 domain calling.** Domains defined by broad H3K9me3 ChIP–seq enrichment across six cell types (HCT116, HFFc6, IMR90, K562, GM12878, H1-hESC) were called using an HMM procedure. H3K9me3 ChIP–seq bigwigs were mean-aggregated at 25 kb, log-transformed and $z$-scored, and binarized with a threshold of 1, and were used to train a two-state Bernoulli HMM using Pomegranate. Smoothed runs of 1 s from the Viterbi parses were used to define domains.

**$P(s)$ curves per IPG.** Scaling curves of contact frequency $P$ as a function of genomic separation $s$ were generated using cooltools by aggregating normalized contact frequency over valid pixels along diagonals of 10-kb-resolution *cis* contact maps limited to IPG domains, with diagonals grouped into geometrically increasing strata of genomic separation. Average contact frequency $P(s)$ curves are displayed using log-log axes.

**Insulation analysis.** Diamond insulation scores[98] were calculated on 25-kb-resolution Hi-C maps with a 100-kb sliding window using the cooltools package. Additionally, an insulation minimum calling

procedure based on peak prominence, described in ref. 97, was used to call insulating loci from the insulation score signal.

**Hi-C pileup maps.** The cooltools package was used to calculate aggregate observed-over-expected contact frequency maps (pileup maps) centered at CTCF sites and bounded by a fixed flanking genomic distance. Pileup maps are centered on the main diagonal at each feature's midpoint.

**Replication timing domain analysis.** To identify early and late replicating domains, a 25-kb binned pandas dataframe was generated using bioframe. HCT116 and DKO replication timing signal tracks were imported into the binned dataframe using pybbi. Missing values were represented as Not a Number (NaN). Domains were identified with a two-state Gaussian HMM using Pomegranate[99]. Viterbi state calls were made on a per bin basis and used for downstream analysis. Neighboring states were merged to create domains then converted to bed files (https://github.com/gspracklin/hmm_bigwigs).

Differential replication timing loci were identified by applying a cutoff of 0.75 on the difference between HCT116 and DKO 50-kb $z$-score tracks. Differentially timed loci separated by up to 250 kb were then merged into larger intervals using bioframe.cluster to produce 199 differentially timed regions.

**Polymer simulations.** Simulations were created using the Polychrom library[100]. The polymer simulations ran using the OpenMM engine for GPU-assisted molecular dynamics simulations[101]. Each simulation modeled 8–11 megabases (Mb) of chromatin fiber as a chain of 1-kb monomers, and included five copies of the system inside the same container. Each simulation was run for 500,000,000 molecular dynamics steps. Periodic boundary conditions were used to maintain a density of 0.2 monomers per cubic nanometer.

The following energies are in terms of $kT$ (the Boltzmann constant times absolute temperature), and distances are measured in terms of the diameter of the monomers, which is 20 nm. Adjacent monomers on the chain are connected by a harmonic bond with potential $U = 100(r - 1)^2$, where $r$ is the distance between the centers of the monomers. Polymer stiffness is modeled by $U = S(1 - \cos(\alpha))$, a force dependent on the angle $\alpha$ formed by three adjacent monomers, and $S$ is a stiffness parameter equal to 1.5.

To model loop extrusion, loop-extruding factors (LEFs) were probabilistically loaded onto the polymer chain at uniformly random positions. Each LEF is represented by a harmonic bond equivalent to the one that connects adjacent monomers on the chain. Each step of one-dimensional (1D) dynamics corresponded to 400 molecular dynamics steps. An LEF with an upstream leg at monomer $i$ will stay at $i$ with probability ½ and move to $i - 1$ with probability ½ each step, unless $i - 1$ is occupied by an LEF or a CTCF. Similarly, a downstream leg at monomer $j$ will stay at $j$ with probability ½ and move to $j + 1$ with probability ½, unless $j + 1$ is occupied by an LEF or CTCF. CTCF sites were placed at fold-change peaks in HCT116 CTCF ChIP–seq (ENCODE ID ENCFF549PGC), with directionality according to CTCF motifs (from ref. 61). Each CTCF had a capture probability of min$((fc - 1)/fc_{med}, 1)$, where fc is the CTCF fold change and $fc_{med}$ is the median CTCF fold change over the region. Legs were released from CTCFs with a probability of 0.006 each monomer step. Each LEF was unloaded with a probability of 1/100 each step of 1D dynamics, and LEFs were separated by an average of 600 monomers.

**Reporting summary**

Further information on research design is available in the Nature Portfolio Reporting Summary linked to this article.

## Data availability

The references and accession numbers of published data used and analyzed in this work are indicated in Supplementary Table 1.

All datasets generated in this study are deposited in the NCBI Gene Expression Omnibus (GEO; http://www.ncbi.nlm.nih.gov/geo/) under the SuperSeries accession number GSE182108; SubSeries for ChIP–seq (GSE182104), Hi-C (GSE182105), Protect-seq (GSE182106) and Repli-seq (GSE182107).

## Code availability

A snakemake workflow for spectral decomposition, clustering and embedding is available at https://github.com/open2c/inspectro. Additional scripts and notebooks used to process the data in our study are available at https://github.com/mirnylab/heterochromatin-paper.

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

## Acknowledgements

We thank P.-O. Estève, A. Goloborodko, G. Edgeworth, E. M. Breville and members of the Dekker and Mirny laboratories for helpful insights and discussion. We thank K. Polovnikov for advice on spectral clustering. We thank N. Fox, J. Gibcus and G. Fudenberg for critical reading and C. Navarro for help editing the manuscript. Schematics in figures (Figs. 4f, 5a, and 7e) were created with BioRender.com. This work was supported by New England Biolabs, Inc. and grants from the National Institutes of Health Common Fund 4D Nucleome Program to J.D. and L.A.M. (grant nos. U54-DK107980 and UM1-HG011536). J.D. is an investigator of the Howard Hughes Medical Institute.

## Author contributions

G.S. and S.P. conceived the study. G.S. designed and performed experiments. N.A., G.S. and M.I. performed data analysis. M.I. and N.C. performed polymer simulations. All authors contributed to data interpretation. G.S., N.A., L.A.M. and J.D. wrote the manuscript.

## Competing interests

The authors declare no competing interests.

## Additional information

**Extended data** is available for this paper at https://doi.org/10.1038/s41594-022-00892-7.

**Correspondence and requests for materials** should be addressed to George Spracklin or Job Dekker.

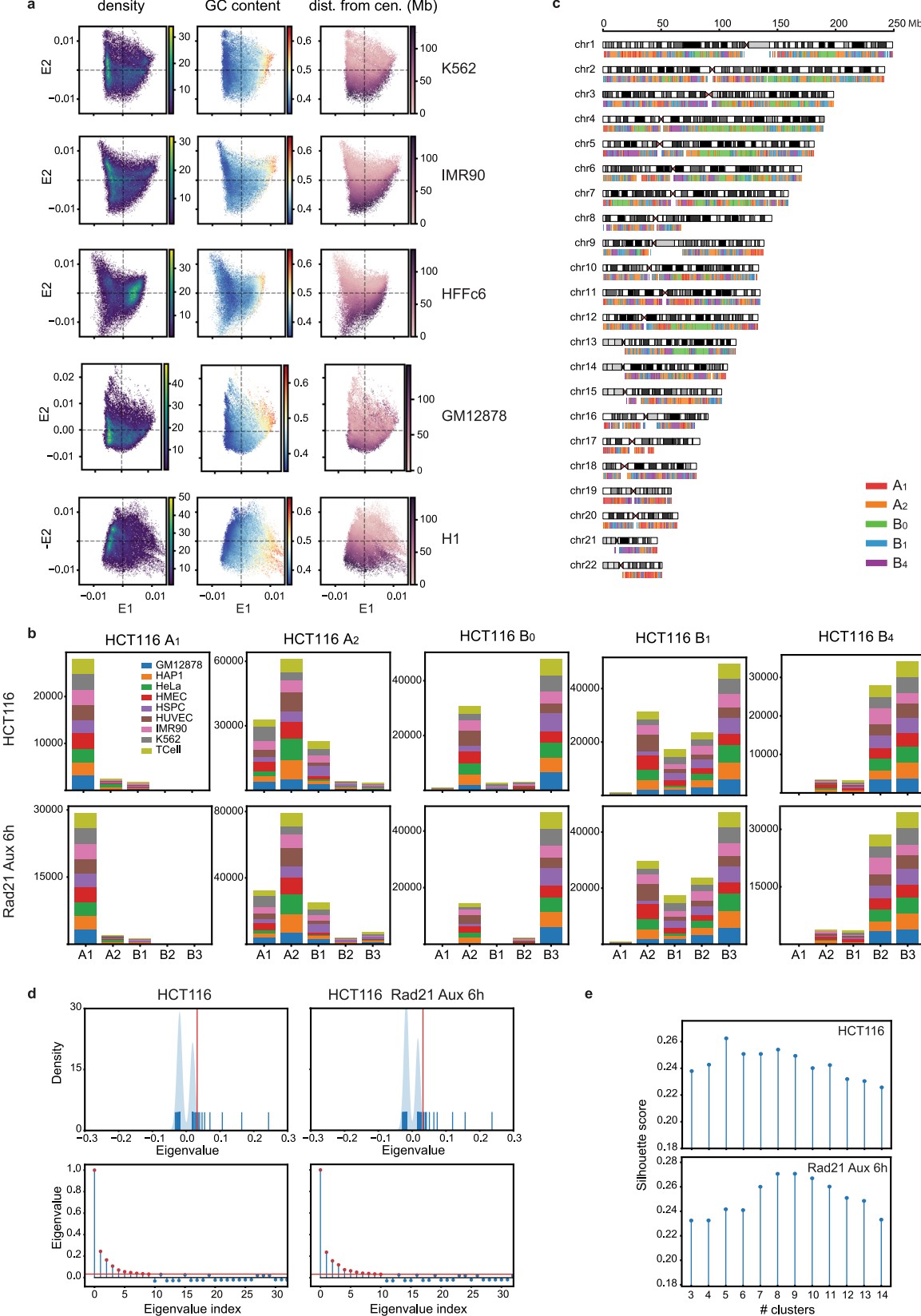

**Extended Data Fig. 1 | Spectral decomposition and clustering in HCT116. (a)** E1 vs. E2 scatter plots of 50 kb genomic bins from five additional cell types (K562, IMR-90, HFFc6, GM12878, H1-hESC) colored by point density (left), GC content (middle), and distance from centromere (right). **(b)** Distributions of SNIPER subcompartment labels assigned to genomic bins in each IPG across nine other cell types for HCT116 (top) and HCT116 RAD21-degron (bottom). **(c)** Ideogram plot of IPGs in HCT116. **(d)** Top, rug plot of the leading 128 eigenvalues for HCT116 (left) and HCT116 RAD21-degron (right). Vertical red line indicates the eigenvalue cutoff. Bottom, same eigenvalues plotted in descending order of absolute value. Eigenvalues corresponding to retained vectors used for clustering are indicated in red. **(e)** Silhouette scores calculated for *k*-means clustering on eigenvectors from HCT116 (top) and HCT116 RAD21-degron (bottom) as a function of the number of clusters, *k*.

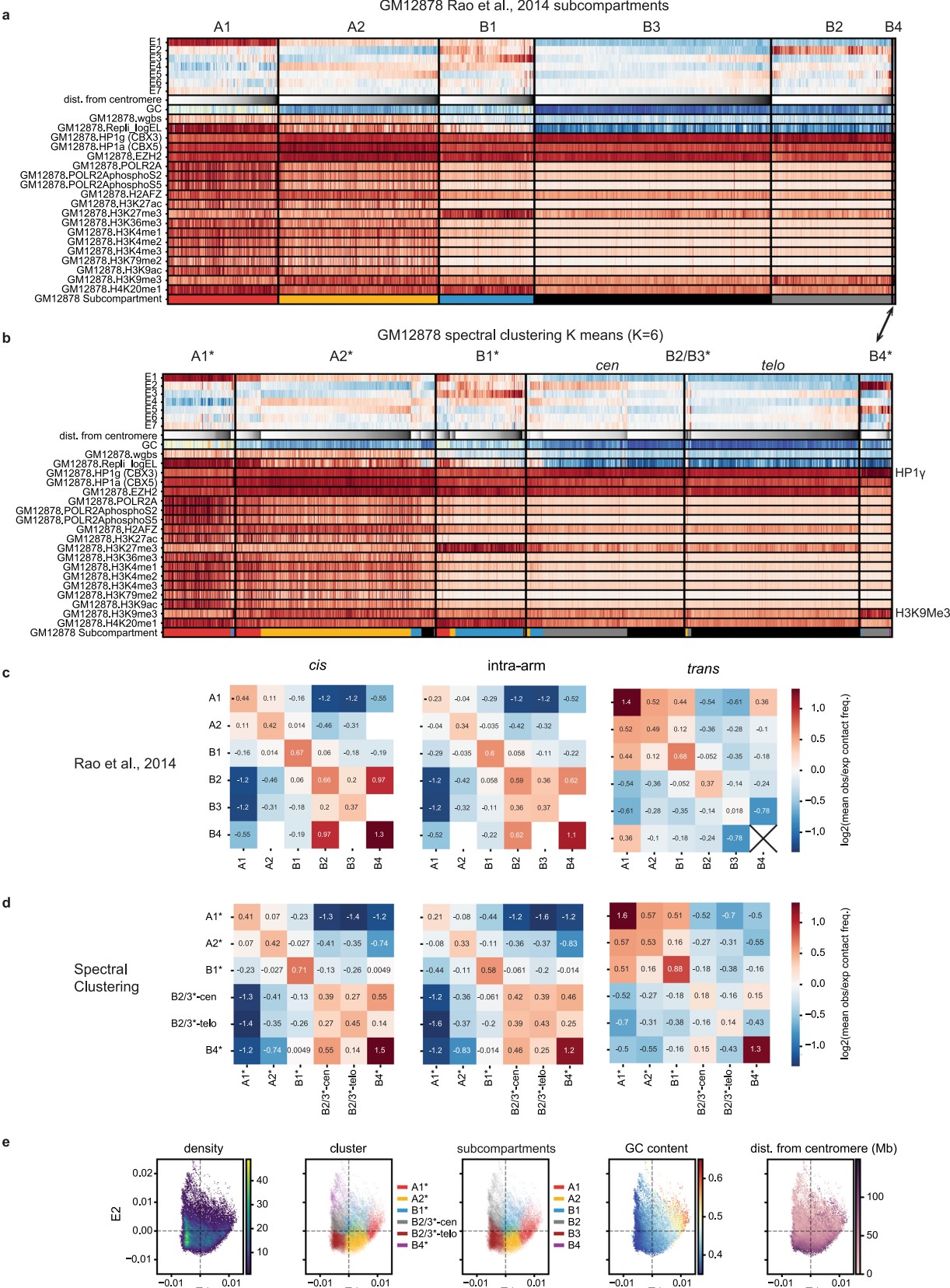

**Extended Data Fig. 2 | See next page for caption.**

**Extended Data Fig. 2 | Spectral decomposition and clustering in GM12878.**
(**a**) Feature heatmap for GM12878 based on 6-subcompartment labels from
(Rao et al.[4]). The tracks displayed are the seven leading eigenvectors (E1-E7), GC
content, fraction CpG methylation, replication timing (Early/Late), and ChIP-seq
for a range of factors and histone modifications. Columns (50-kb bins) within
each subcompartment are sorted by distance from centromere. Colors are
assigned to the subcompartment labels in the last row (A1: red, A2: yellow, B1:
blue, B2: grey, B3: black). (**b**) Feature heatmap for GM12878 based on spectral
clustering of E1-E7 ($k$ = 6). Rows display the same tracks as in (A). Columns within
each cluster are sorted first by subcompartment label assignment, then by
distance from centromere. The last row assigns a color to each bin based on its
subcompartment label as in (A). Names are assigned to the clusters based on
similarity to (A) with addition of an asterisk. The main differences with Rao et
al.[4], subcompartment assignments are (1) a more balanced division between
B2* and B3* based on centromere/telomere proximity and (2) an expanded sixth
cluster, B4*, that acquires B3 loci having highly enriched H3K9me3 and HP1γ.
(**c**) Heatmaps of pairwise mean observed/expected contact frequency between
subcompartments in (Rao et al.[4]) based on *cis* (left), intra-arm (middle), and *trans*
(right) contacts. (**d**) Heatmaps of pairwise mean observed/expected contact
frequency, as in (C), but between spectral clusters from (B). (**e**) E1 vs. E2 scatter
plots from GM12878 colored by point density, GC content, spectral cluster label,
subcompartment label, and distance from centromere.

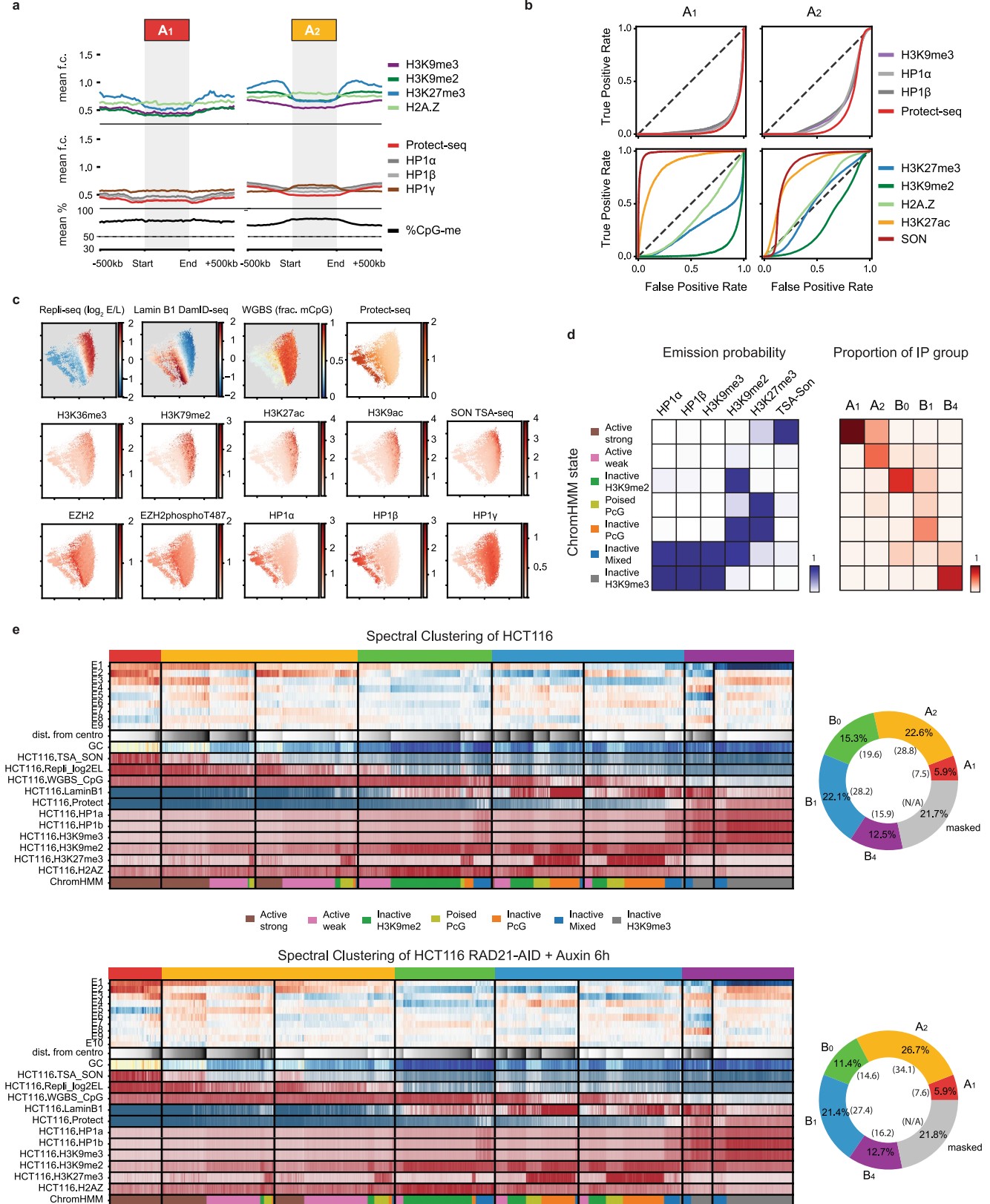

**Extended Data Fig. 3 | See next page for caption.**

**Extended Data Fig. 3 | Chromatin state composition of IPGs in HCT116. (a)** Metaplots displaying signal enrichment for the same features as Fig. 2c for $A_1$ and $A_2$ domains. (**b**) ROC curves assessing the prediction performance of individual 50kb-aggregated functional tracks as binary classifiers as in Fig. 2d but for $A_1$ and $A_2$ loci. Additionally, curves for active marks (ChIP-seq for H3K27ac and TSA-seq for SON) are shown. (**c**) E1 vs. E2 scatter plots of 50-kb bins colored by point density and ChIP-seq for various factors and histone modifications. (**d**) Left, emission probabilities for ChromHMM model on five ChIP-seq for repressive marks and SON (TSA-seq for nuclear speckle marker) trained on 50 kb bins. Right, heatmap showing the distributions of ChromHMM state labels found in each IPG (columns). (**e**) Left, feature heatmaps for spectral clustering on HCT116 (top) and the cohesin-depleted HCT116 RAD21-AID line from (Rao et al.[10]) (bottom).

The tracks displayed are the same as in Fig. 1d but also include various histone marks. Columns (50-kb bins) within each cluster are sorted first by ChromHMM state (as per the model in (D)) and then by distance from centromere. The last row assigns a color to each bin based on its ChromHMM state. When we identify IPGs in Hi-C data from HCT116 cells in which the cohesin subunit RAD21 is depleted, we observe a slight increase in correspondence to ChromHMM labels (Adjusted Rand Index: HCT116 = 0.31, HCT116-RAD21 = 0.35). This is consistent with loop extrusion interfering with innate compartmentalization preferences. Right, donut plots showing hg38 percentage covered by each IPG (top, HCT116; bottom, HCT116 RAD21-AID). Note: translocations and unmappable areas are masked. Percentages excluding translocations and unmappable areas are in parentheses.

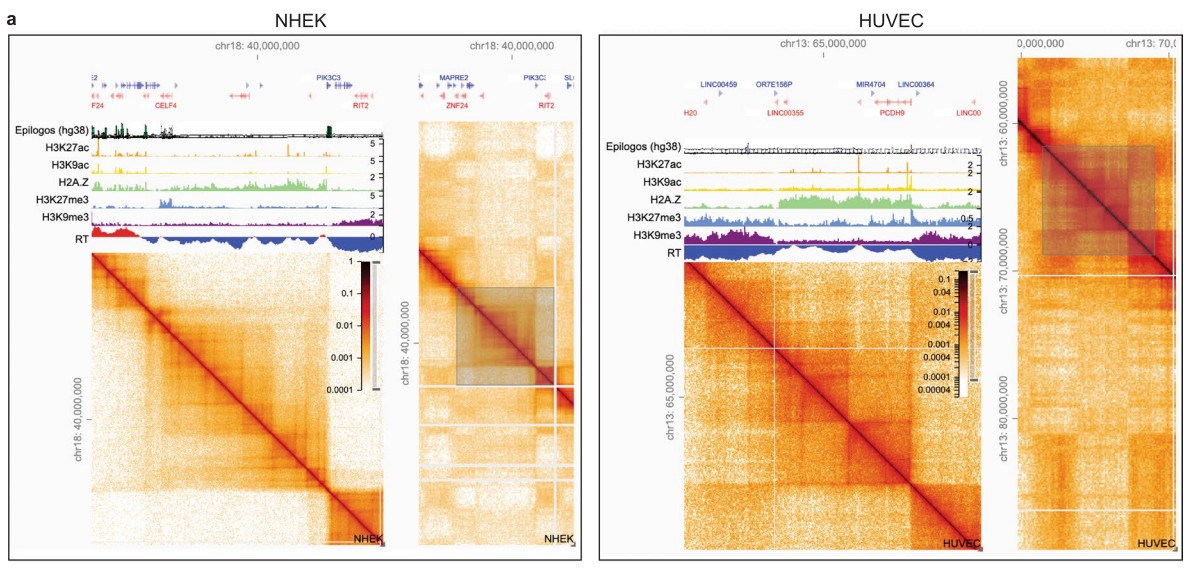

**Extended Data Fig. 4 | Examples of B₀-like domain in other cell types and mixed-state domains (constitutive/poised) in HCT116.** (**a**) Examples of B₀-like domains in NHEK (left) and HUVEC (right) cells. HiGlass views display an overview panel on the right with a corresponding detailed zoom-in on the left including ChIP-seq and two-stage Repli-seq tracks. (**b**) Two example regions that exhibit a ChromHMM state, Inactive Mixed, that emits a combination of H3K9me3-HP1α/β (similar to B₄) and H3K9me2 (similar to B₀) (see Extended Data Fig. 3) display long-range Hi-C profiles that appear to be a superposition of B₀ and B₄. Highlighted boxes further illustrate continuous domains with fractional heights relative to neighboring domains in Protect-seq, ChIP-seq, and WGBS, suggestive of population heterogeneity or allelic imbalance. Note the faint appearance of loop extrusion features in the Hi-C maps as well.

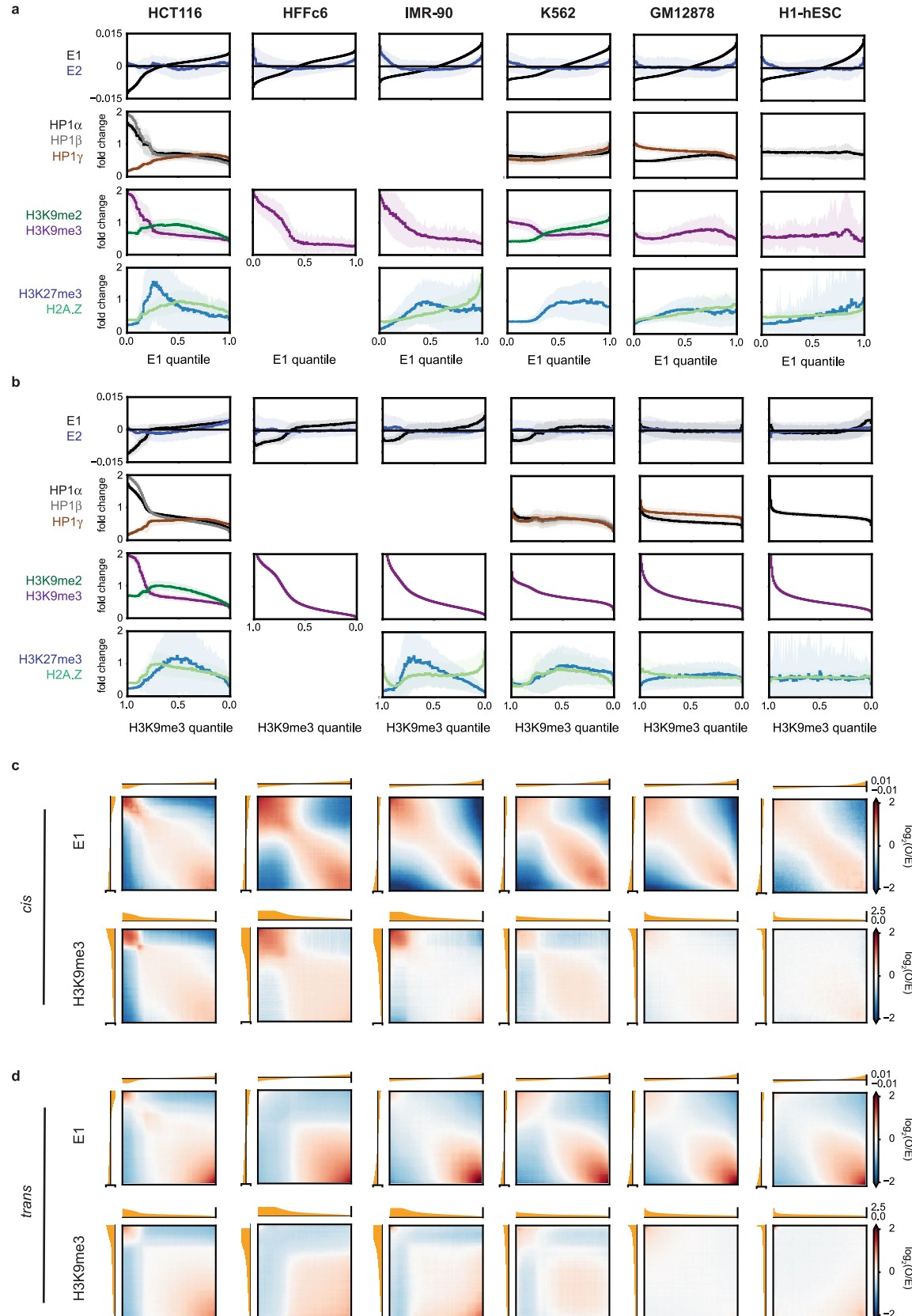

**Extended Data Fig. 5 | See next page for caption.**

**Extended Data Fig. 5 | Comparative analysis of compartmentalization and heterochromatin marks.** Comparative analysis of genome organization and heterochromatic marks across HCT116, HFFc6, IMR90, K562, GM12878 and H1-hESC. (**a**) Histograms of ChIP-seq signal for repressive histone marks as in Fig. 3a based on eigenvector (E1) percentile and displayed in ascending order of E1 rank. Includes additional histograms for E1 and E2 (top) and data for two additional cell types: lung fibroblasts IMR-90 and foreskin fibroblasts HFFc6. (**b**) Histograms of ChIP-seq signal for repressive histone marks as in Fig. 3d based on H3K9me3 percentile and displayed in descending order of H3K9me3 rank. Includes additional histograms for E1 and E2 (top) and data for IMR-90 and HFFc6. (**c**) Bivariate summary maps of *cis* observed/expected contact frequency as in Fig. 3b, c based on E1 percentile in ascending order (top) and H3K9me3 percentile in descending order (bottom). (**d**) Bivariate summary maps as in (C) but describing observed/expected contact frequency in *trans*. In K562, GM12878 and H1 cells loci with low/negative E1 values still prefer to interact with other loci with similar E1 values even though in these cells most of these loci do not display strong H3K9me3-HP1 enrichment.

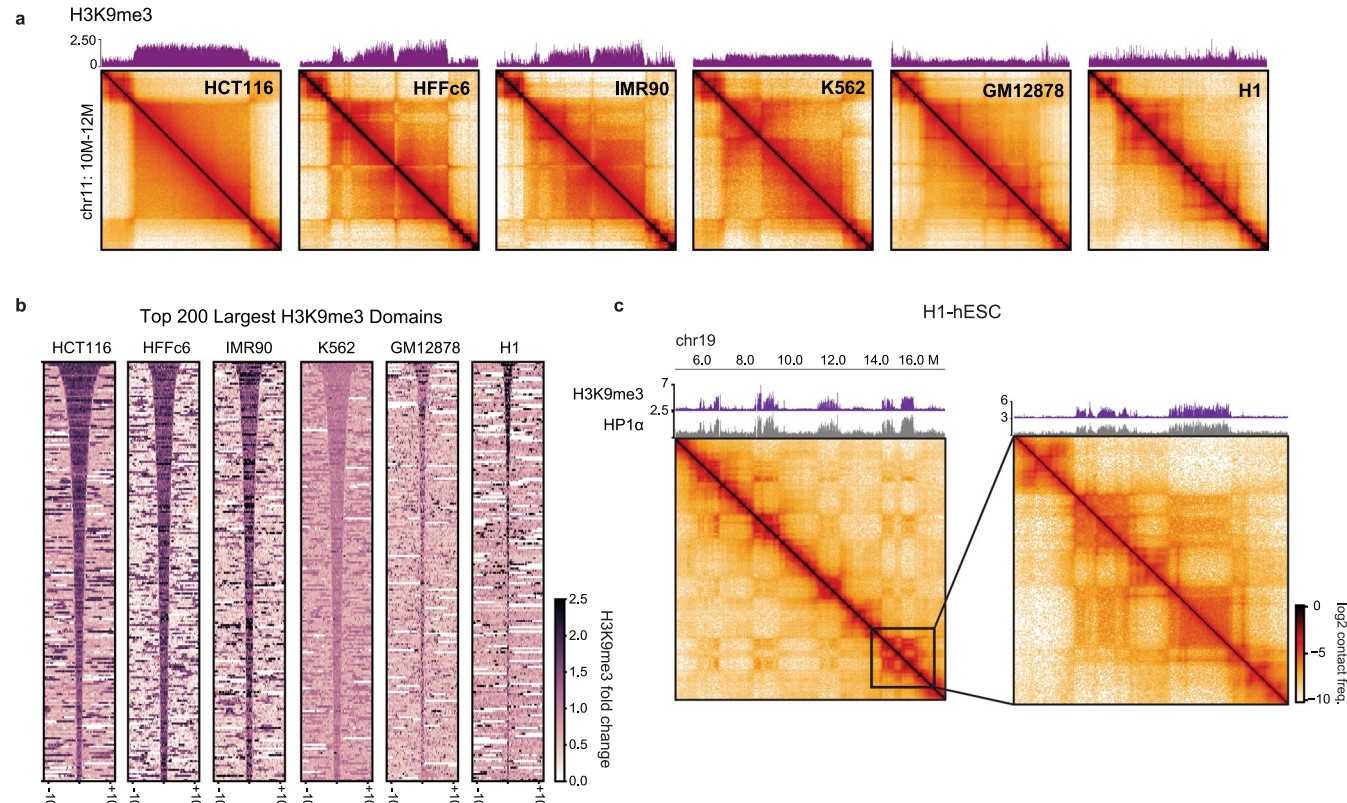

**Extended Data Fig. 6 | Comparative analysis of H3K9me3 domains.**
Comparative analysis of genome organization and heterochromatic marks across HCT116, HFFc6, IMR90, K562, GM12878 and H1-hESC. (**a**) Expanded example domain across cell types as in Fig. 3e including data for IMR-90 and HFFc6. (**b**) Stacked signal heatmaps of H3K9me3 signal centered at the top 200 largest H3K9me3 domains detected in six cell types. (**c**) Example of homotypic interactions at H3K9me3-HP1α domains on chr19 in H1-hESC.

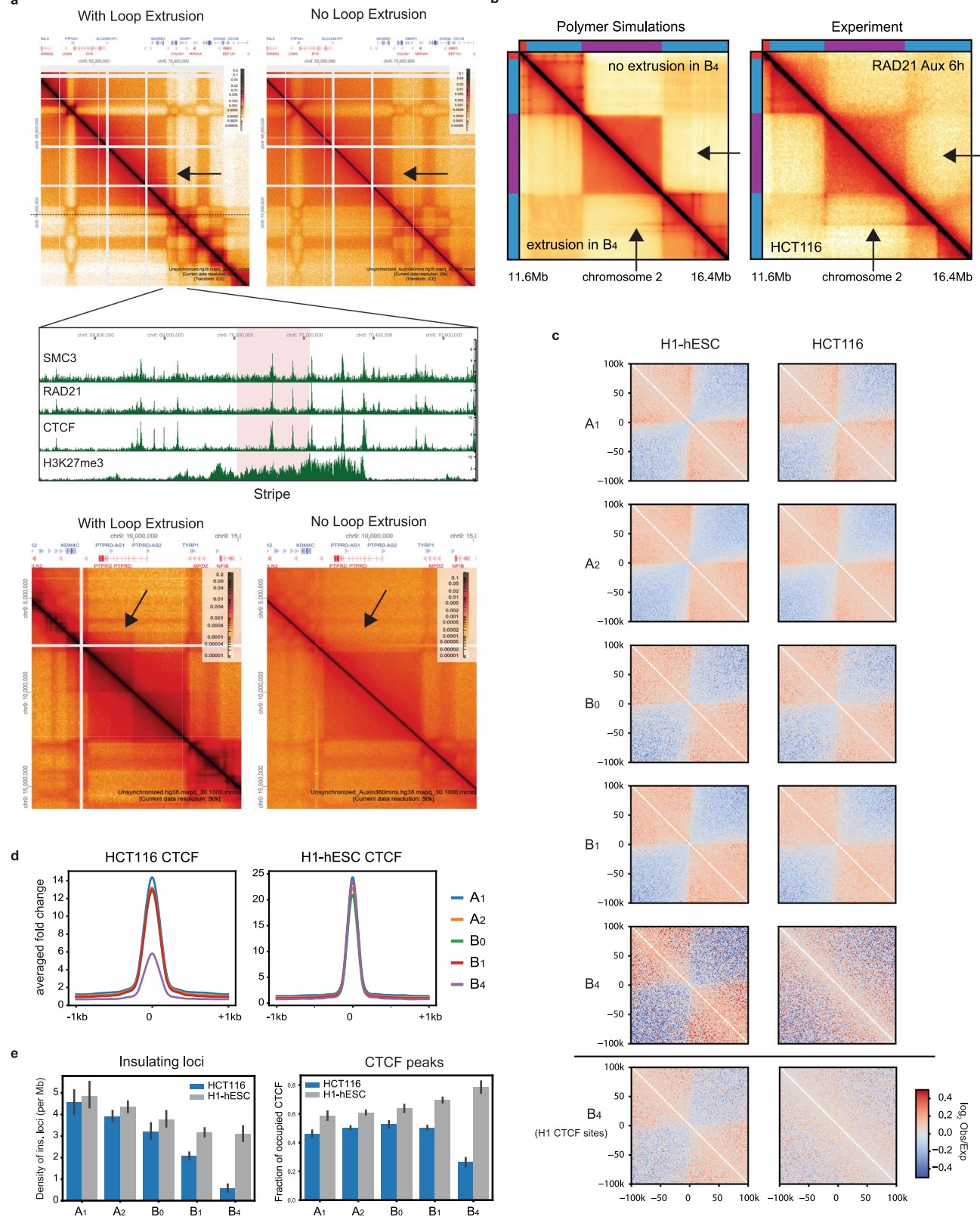

**Extended Data Fig. 7 | See next page for caption.**

**Extended Data Fig. 7 | Evidence of loop extrusion but lack of CTCF within heterochromatin domains.** (**a**) Two examples of cohesin-dependent loop extrusion features traversing a $B_4$ domain. Hi-C maps of HCT116 (left columns) and HCT116-RAD21 auxin depletion (right columns). Arrows indicate loop extrusion features that are dependent on the cohesin complex: stripe (top Hi-C map) and TAD (bottom Hi-C map). Middle panel, ChIP-seq tracks of SMC3, RAD21, CTCF, and H3K27me3 for the stripe (highlighted in pink) and surrounding region (**b**) Contact frequency maps from *in silico* polymer simulations (left) compared to experimental Hi-C (right). Arrows indicate a stripe next to a $B_4$ domain that extends parallel to its edge in HCT116. Experimental data is replicated when cohesin traversal is permitted (lower triangle) and does not appear when loop extrusion is blocked at the $B_4$ domain (upper triangle). (**c**) Average observed/expected maps from HCT116 and H1-hESC Hi-C maps centered at HCT116 CTCF binding sites within each HCT116-defined IPG. Bottom row, same for $B_4$ but using H1-hESC CTCF binding sites. Expected maps are calculated separately for each IPG. (**d**) Average fold enrichment of CTCF ChIP-seq across all known CTCF sites used in (C) for HCT116 and H1-hESC. (**e**) Left, density of insulating loci in H1-hESC (4DNFIGDQ72ID) and HCT116 (4DNFIBKY9EG9) cells grouped by HCT116 IPG. Bar heights give the mean CTCF density across all domains belonging to an IPG in each cell type. Error bars represent 95% confidence intervals. $n(A_1) = 310$, $n(A_2) = 1448$, $n(B_0) = 711$, $n(B_1) = 1333$, $n(B_4) = 382$ domains from each IPG. Right, fraction of peaks detected at all known CTCF sites (from Maurano et al.[61]) occupied in HCT116 (ENCFF171SNH) and H1-hESC (ENCFF692RPA) ChIP-seq grouped by HCT116 IPG. Bars represent the fraction of (Maurano et al.[61]) CTCF sites occupied per domain in each cell type. Bar heights give the mean fraction occupied across all domains belonging to an IPG. Error bars and n values as on the left.

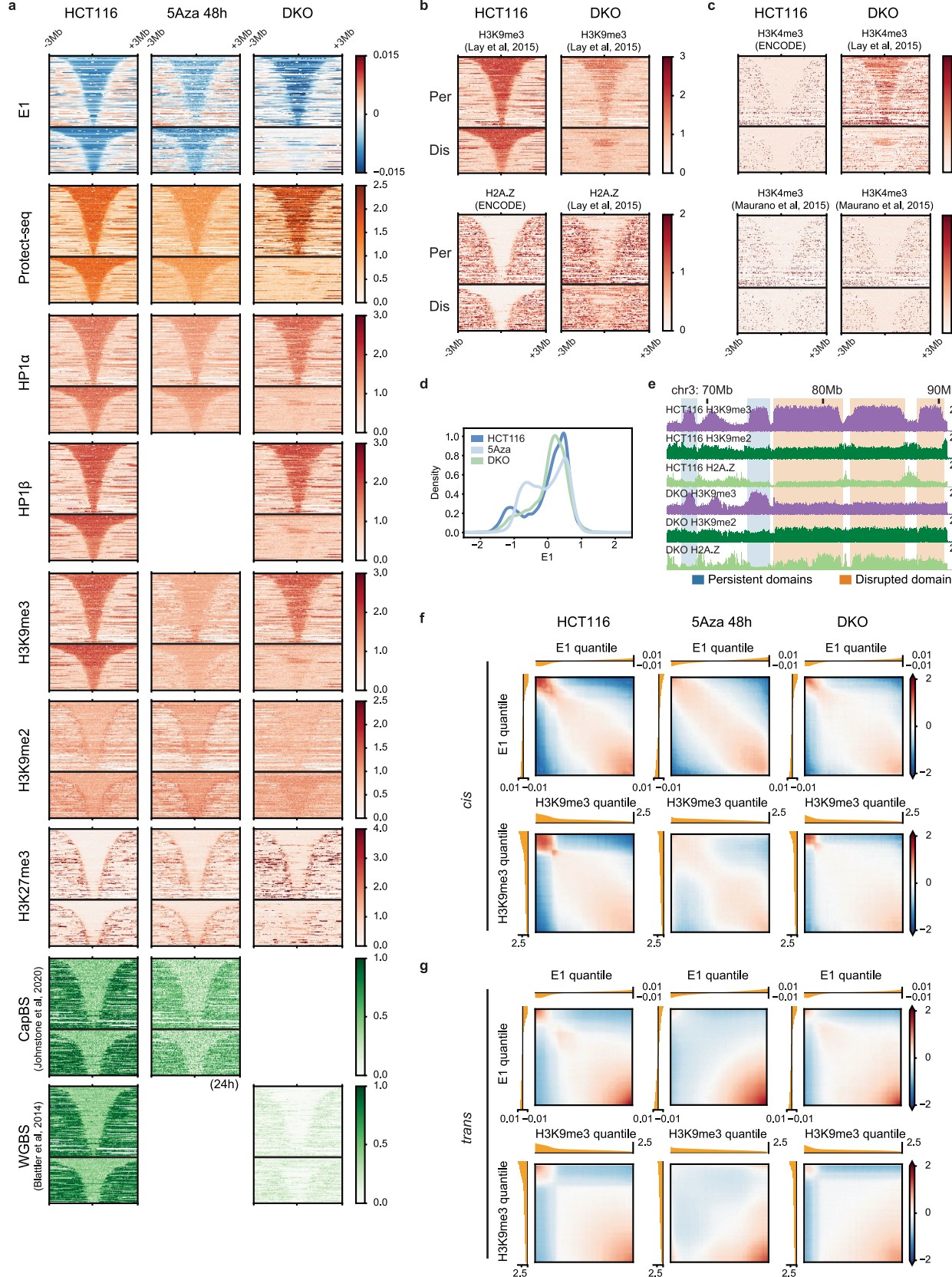

**Extended Data Fig. 8 | See next page for caption.**

**Extended Data Fig. 8 | Maintenance of H3K9me3-HP1α/β heterochromatin depends on DNA methylation homeostasis.** (**a**) Stacked signal heatmaps centered at persistent and disrupted $B_4$ domains (not scaled) displaying various signal tracks in HCT116, 5Aza-treated cells, and DKO. Hybrid Selection Capture BS DNA methylation (CapBS) data were obtained from (Johnstone et al.[77]) and WGBS data were obtained from (Blattler et al.[76]). (**b**) Stacked signal heatmaps centered at persistent and disrupted $B_4$ domains identified in this study displaying H3K9me3 and H2A.Z signal from (Lay et al.[55]). Note that the H3K9me3 domains in the DKO line used in that study appear slightly divergent from those detected here. (**c**) Stacked signal heatmaps similar to (B) but displaying H3K4me3 ChIP-

seq from (Lay et al.[55]) and (Maurano et al.[61]). The first study shows a remarkable DKO-specific co-enrichment of H3K4me3 signal with H3K9me3 marking persistent domains, but this result was not reproduced in (Maurano et al.[61]). (**d**) KDE plots of E1 signal in HCT116, 5Aza-treated cells, and DKO. (**e**) Example region (chr3:70–90 Mb) showing persistent (blue shading) and disrupted (orange shading) domains. ChIP-seq tracks for H3K9me2, H3K9me3, and H2A.Z in HCT116 (top 3 tracks) and DKO (bottom 3 tracks) (**f**) Bivariate summary maps of *cis* observed/expected contact frequency based on E1 percentile (top) and H3K9me3 percentile (bottom) in HCT116, 5Aza-treated cells, and DKO. (**g**) Same as (F) but for *trans* contact frequency in HCT116, 5Aza-treated cells, and DKO.

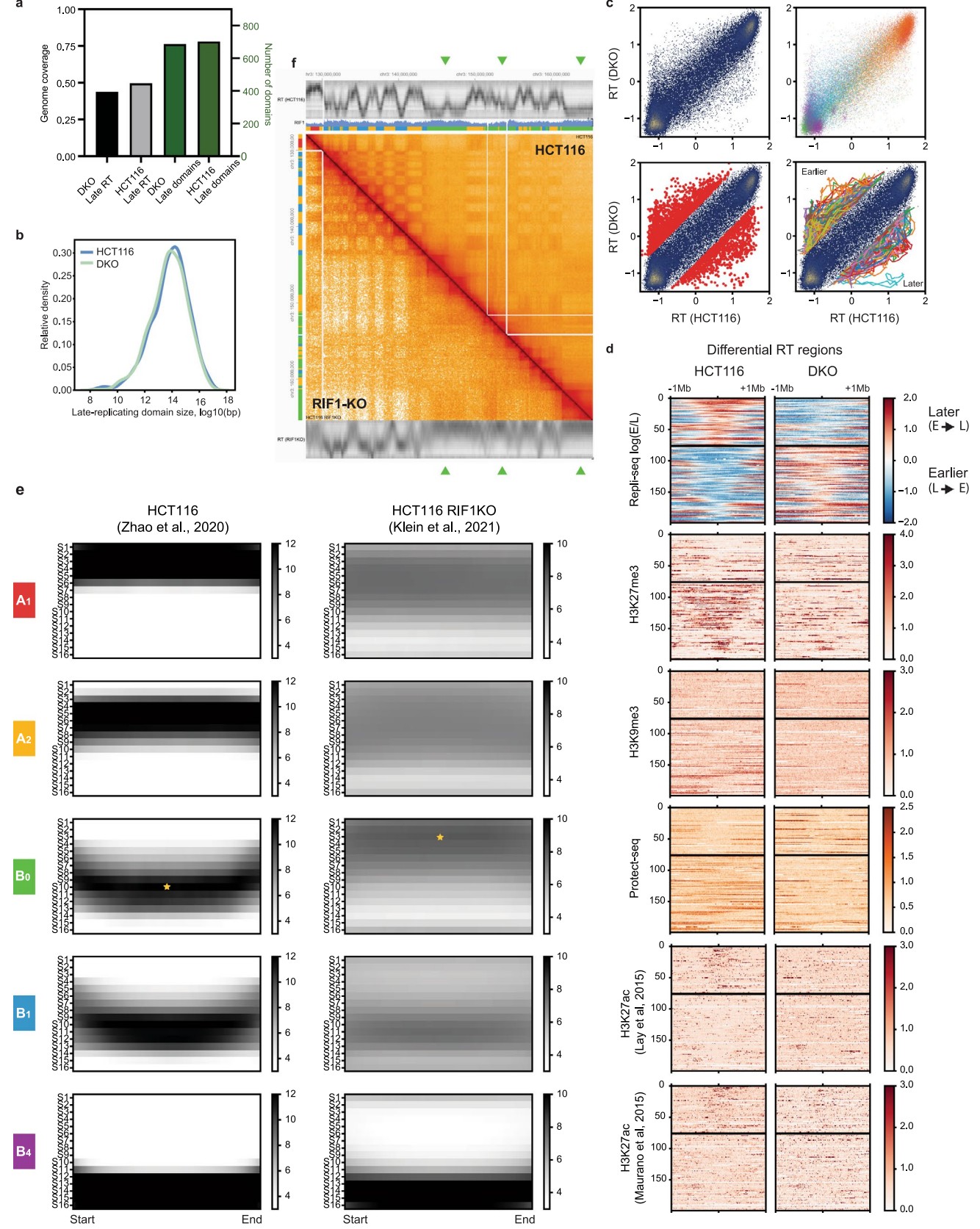

**Extended Data Fig. 9 | See next page for caption.**

**Extended Data Fig. 9 | Late replication timing of $B_0$ in HCT116 is RIF1-dependent, while shifts to earlier RT in DKO are associated with loss of H3K27me3.** (**a**) Total number (green) and genome coverage (black) of late replicating domains detected in HCT116 and DKO using a Gaussian HMM. (**b**) KDE plots of domain size of late replicating domains (log10) in HCT116 and DKO. (**c**) Differential replication timing analysis. Top: Left, scatter plot of 50-kb genomic bins based on z-scored Repli-seq log2(Early/Late) in HCT116 vs DKO. Right, same scatter plot colored by IPG label. Bottom: Left, same scatter plot with loci exhibiting a change >= 0.75 highlighted in red. Right, same scatter plot with continuous merged differential regions connected using colored lines. (**d**) Stacked signal heatmaps centered at differentially replicating regions (not scaled) divided into later/delayed onset (top) and earlier/hastened onset (bottom) regions displaying various signal tracks in HCT116 and DKO cells (n = 199). (**e**) Aggregate heatmaps of 16-stage Repli-seq from HCT116 (Zhao et al.[60]) and HCT116 RIF1-KO (Klein et al.[56]) derived from uniformly scaled IPG domains. Star icons indicate the modal stage in $B_0$ domains: S10 in HCT116 and S3 in HCT116 RIF1-KO. (**f**) HiGlass view of 16-stage Repli-seq and Hi-C for HCT116 (top right) and RIF1-KO (bottom left). Three $B_0$ regions that shift replication timing from late to early in RIF1-KO are denoted with green arrow heads.

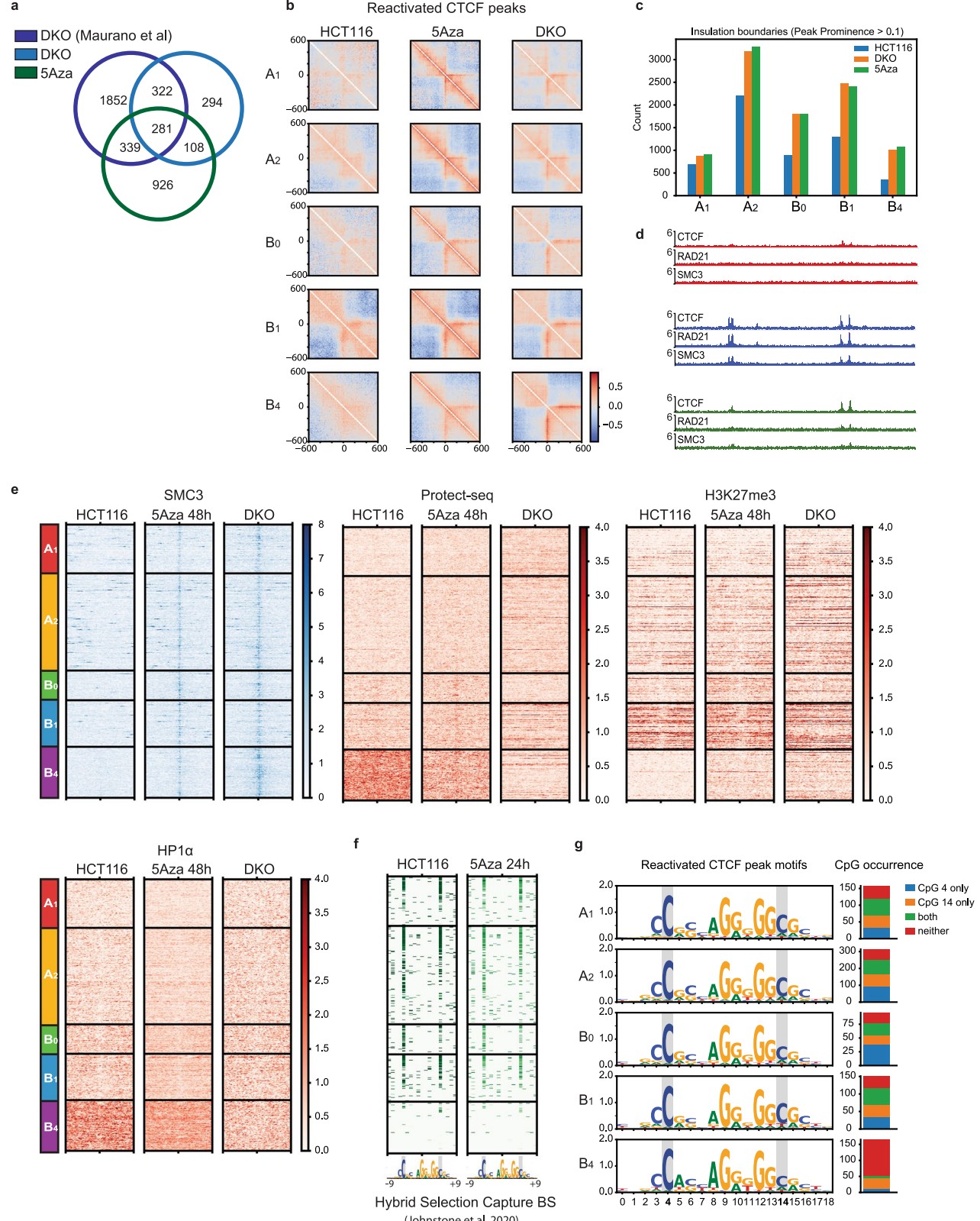

**Extended Data Fig. 10 | See next page for caption.**

**Extended Data Fig. 10 | Reactivated CTCF sites.** (**a**) Venn diagram of CTCF peaks in DKO (this study), 5Aza (this study), and DKO (Maurano et al.[61]). Union between CTCF peaks used to define reactivated CTCF sites. (**b**) Average observed/expected Hi-C maps around reactivated CTCF binding sites within each IPG centered at CTCF motifs oriented as indicated in HCT116 (left), 5Aza (center), and DKO (right) cells. (**c**) Quantification of total number of insulating loci with peak prominence score > 0.1 per IPG. (**d**) Example region (chr11:39–40 Mb) of reactivated CTCF sites blocking cohesin (RAD21 and SMC3). (**e**) Stacked heatmaps of reactivated CTCF sites for HCT116, 5Aza, and DKO cells centered on the CTCF motif displaying ChIP-seq signal for SMC3 (upper left), Protect-seq (middle), H3K27me3 (right),

and HP1α (lower left) flanked by ±5 kb and segregated by IPG. (**f**) Similar to Fig. 7c. Stacked heatmaps around reactivated CTCF site core motifs (19 bp) for HCT116 and 5Aza-treated cells displaying fraction CpG methylation using hybrid selection capture bisulfite sequencing data from (Johnstone et al.[77]). (**g**) Left: sequence logos for the reactivated CTCF motifs in each IPG. Right: frequencies of CpG occurrence at motif positions 4 and 14 in each set of reactivated CTCF sites. Note: nucleotides 4 and 14 depend on the motif start, other publications refer to these CpG nucleotides as 2 and 12 (for example Hashimoto et al.[63]) or 1 and 11 (for example Wang et al.[33]).

# Reporting Summary

## Statistics

For all statistical analyses, confirm that the following items are present in the figure legend, table legend, main text, or Methods section.

| n/a | Confirmed | |
|---|---|---|
| ☐ | ☒ | The exact sample size (*n*) for each experimental group/condition, given as a discrete number and unit of measurement |
| ☐ | ☒ | A statement on whether measurements were taken from distinct samples or whether the same sample was measured repeatedly |
| ☒ | ☐ | The statistical test(s) used AND whether they are one- or two-sided *Only common tests should be described solely by name; describe more complex techniques in the Methods section.* |
| ☒ | ☐ | A description of all covariates tested |
| ☒ | ☐ | A description of any assumptions or corrections, such as tests of normality and adjustment for multiple comparisons |
| ☐ | ☒ | A full description of the statistical parameters including central tendency (e.g. means) or other basic estimates (e.g. regression coefficient) AND variation (e.g. standard deviation) or associated estimates of uncertainty (e.g. confidence intervals) |
| ☒ | ☐ | For null hypothesis testing, the test statistic (e.g. *F*, *t*, *r*) with confidence intervals, effect sizes, degrees of freedom and *P* value noted *Give P values as exact values whenever suitable.* |
| ☒ | ☐ | For Bayesian analysis, information on the choice of priors and Markov chain Monte Carlo settings |
| ☒ | ☐ | For hierarchical and complex designs, identification of the appropriate level for tests and full reporting of outcomes |
| ☒ | ☐ | Estimates of effect sizes (e.g. Cohen's *d*, Pearson's *r*), indicating how they were calculated |

*Our web collection on statistics for biologists contains articles on many of the points above.*

## Software and code

Policy information about availability of computer code

| Data collection | Illumina Nextseq 500 was used for DNA sequencing. SONY-SH800 Software was used for flow cytometry. |
|---|---|
| Data analysis | Custom scripts and notebooks are available at https://github.com/mirnylab/heterochromatin-paper. The spectral clustering pipeline is available at https://github.com/open2c/inspectro. macs2 (v2.2.7.1), UCSC tools 357, samtools (v1.7), bedtools (v2.29.2), python (v3.8.5), snakemake (v5.31.1), bwa (v0.7.17), phantompeakqualtools (v1.2.2), bioframe (v0.3.3), crossmap (v0.6.4), pybbi (v0.3.2), cooltools (v0.5.1), cooler (v0.8.11), distiller (v0.3.3), gimmemotifs (0.17.1), picard (v2.27.4), scikit-learn (v1.1.1), UMAP (v0.5.3), matplotlib (v3.5.2), datashader (v0.14.1), chromHMM (v1.23), plotly (v5.9.0), pomegranate (v0.4.0), polychrom (v0.1.0), |

For manuscripts utilizing custom algorithms or software that are central to the research but not yet described in published literature, software must be made available to editors and reviewers. We strongly encourage code deposition in a community repository (e.g. GitHub). See the Nature Portfolio guidelines for submitting code & software for further information.

## Data

Policy information about availability of data

All manuscripts must include a data availability statement. This statement should provide the following information, where applicable:
- Accession codes, unique identifiers, or web links for publicly available datasets
- A description of any restrictions on data availability
- For clinical datasets or third party data, please ensure that the statement adheres to our policy

The references and accession numbers of published data used and analyzed in this work are indicated in Supplementary Table 2. All data sets generated in this study are deposited in the NCBI Gene Expression Omnibus (GEO; http://www.ncbi.nlm.nih.gov/geo/) under the SuperSeries accession number GSE182108. SubSeries for ChIP-seq (GSE182104), Hi-C (GSE182105), Protect-seq (GSE182106), and Repli-seq (GSE182107).

# Field-specific reporting

Please select the one below that is the best fit for your research. If you are not sure, read the appropriate sections before making your selection.

☒ Life sciences          ☐ Behavioural & social sciences          ☐ Ecological, evolutionary & environmental sciences

For a reference copy of the document with all sections, see nature.com/documents/nr-reporting-summary-flat.pdf

# Life sciences study design

All studies must disclose on these points even when the disclosure is negative.

| | |
|---|---|
| Sample size | No sample size calculations were made. All quantitative data shown is an average of at least 2 independent experiments depending on the assay as noted in Supplemental Table 1. Experiments were performed in bulk and each replicate represents millions of cells. |
| Data exclusions | No data were excluded from the analysis, except for Hi-C normalization. As recommended for the analysis of Hi-C data, we removed bins with low-coverage as well as contact signals between same of adjacent genomic bins. |
| Replication | All attempts to replicate experiments succeeded. All experiments were performed in at least duplicate. |
| Randomization | No randomization was done. Experiments were based on the genotype and/or condition being tested and performed in parallel. |
| Blinding | Investigators were not blinded to group allocation during data collection and/or analysis. Experiments were performed in parallel. |

# Reporting for specific materials, systems and methods

We require information from authors about some types of materials, experimental systems and methods used in many studies. Here, indicate whether each material, system or method listed is relevant to your study. If you are not sure if a list item applies to your research, read the appropriate section before selecting a response.

### Materials & experimental systems

| n/a | Involved in the study |
|---|---|
| ☐ | ☒ Antibodies |
| ☐ | ☒ Eukaryotic cell lines |
| ☒ | ☐ Palaeontology and archaeology |
| ☒ | ☐ Animals and other organisms |
| ☒ | ☐ Human research participants |
| ☒ | ☐ Clinical data |
| ☒ | ☐ Dual use research of concern |

### Methods

| n/a | Involved in the study |
|---|---|
| ☐ | ☒ ChIP-seq |
| ☐ | ☒ Flow cytometry |
| ☒ | ☐ MRI-based neuroimaging |

## Antibodies

| | |
|---|---|
| Antibodies used | anti-BrdU (BD:555627), anti-CTCF (CST:#3418), anti-RAD21 (abcam:ab992), anti-SMC3 (abcam:ab9263), anti-H3K27me3 (CST:#9733), anti-H3K9me2 (CST:#9753), anti-H3K9me3 (EMD:07-442), anti-HP1alpha (CST:2616) |
| Validation | Antibodies used in this study are commercially available and have been validated by the manufacturer. CTCF, H3K27me3, H3K9me2, HP1alpha antibodies were validated for ChIP-seq by Cell Signaling Technology. When possible datasets generated for this study were compared with those from the ENCODE project. |

## Eukaryotic cell lines

Policy information about cell lines

| | |
|---|---|
| Cell line source(s) | HCT116 (ATCC CCL-247)<br>DKO (HCT116 derived, gift from B. Vogelstein to S.Pradhan, see ref. Rhee 2002) |
| Authentication | None of the cell lines have been authenticated. |
| Mycoplasma contamination | Cell lines were not tested for mycoplasma contamination but no indication of contamination was observed. |
| Commonly misidentified lines<br>(See ICLAC register) | HCT116 is in the ICLAC registry. We used HCT116 verified by the ATCC because it is a well established, near-diploid, and has many available epigenetic datasets. |

# ChIP-seq

## Data deposition

☒ Confirm that both raw and final processed data have been deposited in a public database such as GEO.

☒ Confirm that you have deposited or provided access to graph files (e.g. BED files) for the called peaks.

| | |
|---|---|
| Data access links *May remain private before publication.* | https://www.ncbi.nlm.nih.gov/geo/query/acc.cgi?acc=GSE182108 |
| Files in database submission | See Supplemental Table 1. |
| Genome browser session (e.g. UCSC) | HCT116 with interaction profile groups: https://resgen.io/viewer/S_dmDHLJQp66b8N9uMFQ0Q Reactivated CTCF and Hi-C map: https://resgen.io/l/?d=I5UAnlnWREWRTqfvl2y5Zw Compartment defect: https://resgen.io/l/?d=cd9AchRDQze6P_tuR-F4Iw |

## Methodology

| | |
|---|---|
| Replicates | All ChIP-seq experiments were performed in at least 2 independent experiments depending on the assay as noted in Supplemental Table 1. |
| Sequencing depth | All replicate ChIP-seq experiments were sequenced to a depth >10M as a single-end read. |
| Antibodies | anti-BrdU (BD:555627), anti-CTCF (CST:#3418), anti-RAD21 (abcam:ab992), anti-SMC3 (abcam:ab9263), anti-H3K27me3 (CST:#9733), anti-H3K9me2 (CST:#9753), anti-H3K9me3 (EMD:07-442), anti-HP1alpha (CST:2616) |
| Peak calling parameters | All ChIP-seq data, including data from (Lay et al., 2015) and (Maurano et al., 2015) but excluding those obtained from the ENCODE portal, were processed following the steps of the ENCODE ChIP-seq pipeline (https://github.com/ENCODE-DCC/chip-seq-pipeline2) with slight modifications using a simplified custom snakemake workflow. Briefly, reads were mapped to hg38 using bwa mem (Li, 2013). Alignment files (BAM format) were filtered for quality and duplicates using the samtools (Li et al., 2009). Cross-correlation analysis and fragment length estimation for single-ended datasets was performed using the phantompeakqualtools package (Landt et al., 2012). Signal track (target over input) and peaks were generation was performed using MACS2 default parameters (Y. Zhang et al., 2008). |
| Data quality | Raw read files were assessed using fastqc prior to processing. Cross-correlation and phantom peak analysis was used to ensure data quality and estimate fragment lengths. All datasets used for peak calling received a quality tag of 1 (High) or 2 (veryHigh) from cross-correlation analysis using phantompeakqualtools. |
| Software | macs2 (v2.2.7.1), UCSC tools 357, samtools (v1.7), bedtools (v2.29.2), python (v3.8.5), snakemake (v5.31.1), bwa (v0.7.17), phantompeakqualtools (v1.2.2) |

# Flow Cytometry

## Plots

Confirm that:

☒ The axis labels state the marker and fluorochrome used (e.g. CD4-FITC).

☒ The axis scales are clearly visible. Include numbers along axes only for bottom left plot of group (a 'group' is an analysis of identical markers).

☒ All plots are contour plots with outliers or pseudocolor plots.

☒ A numerical value for number of cells or percentage (with statistics) is provided.

## Methodology

| | |
|---|---|
| Sample preparation | Repli-seq was performed and analyzed as described in (Marchal et al., 2018). HCT116 cells were pulsed with 100uM BrdU for 2hrs, trypsinized, ethanol fixed, stained with propidium iodide and FACS sorted (SONY SH-800) based on DNA content (early S v late S). |
| Instrument | SONY-SH800 |
| Software | SONY-SH800 Software |
| Cell population abundance | During two-stage Repli-seq cells are sorted based on DNA content (i.e. propidium iodide intensity). Due to the nature of unsynchronized cell populations and the imprecision of DNA content as a marker for early/late replication no further quality controls can be performed. |

Gating strategy | Two gates were used to split cells in S-phase. G1 and G2 are easily identifiable and the gates were constructed to split G1/G2 evenly as the early and late fractions.

☒ Tick this box to confirm that a figure exemplifying the gating strategy is provided in the Supplementary Information.

