## [Peer Review File · Nature Structural & Molecular Biology]

Peer Review Information

Manuscript Title: Diverse silent chromatin states modulate genome compartmentalization and loop extrusion barriers

Corresponding author name(s): Job Dekker, George Spracklin

Editorial Notes:

Redactions – transferred manuscripts (mention of previous referee reports from elsewhere) This manuscript has been previously reviewed at another journal. This document only contains reviewer comments, rebuttal and decision letters for versions considered at Nature Structural & Molecular Biology. Mentions of prior referee reports have been redacted

Reviewer Comments & Decisions:

Decision Letter, first revision:

Message: Our ref: NSMB-A46086-T

8th Jun 2022

Dear Dr. Dekker,

Thank you for submitting your revised manuscript "Heterochromatin diversity modulates genome compartmentalization and loop extrusion barriers" (NSMB-A46086-T). It has now been seen by the original referees and their comments are below. The reviewers find that the paper has improved in revision, and therefore we'll be happy in principle to publish it in Nature Structural & Molecular Biology, pending minor revisions to satisfy the referees' final requests and to comply with our editorial and formatting guidelines.

To facilitate our work at this stage, we would appreciate if you could send us the main text as a word file. Please make sure to copy the NSMB account (cc'ed above).

Sincerely,

Carolina

Carolina Perdigoto, PhD
Chief Editor
Nature Structural & Molecular Biology
orcid.org/0000-0002-5783-7106

Reviewer #1 (Remarks to the Author):

The revised version of the manuscript has improved clarity and addresses many of my previous comments. I agree that some of the experiments I originally suggested would need a stretch of work that might be kept for future work and I am in favour of publication of the current version of the manuscript.

Reviewer #2 (Remarks to the Author):

The authors have addressed all the points that I raised and I don't have further concerns about the work.

Reviewer #3 (Remarks to the Author):

The authors have responded to the remarks this reviewer has raised for the first submission of this manuscript [Redacted]. They either revising the manuscript or convincingly discussing why certain analyses were not possible. I find it acceptable for publication in Nature Structural & Molecular Biology, which is in my opinion a very suitable journal for this manuscript.

Decision Letter, author guidance:

Message: Our ref: NSMB-A46086-T

5th Jul 2022

Dear Dr. Dekker,

Thank you for your patience as we've prepared the guidelines for final submission of your Nature Structural & Molecular Biology manuscript, "Heterochromatin diversity modulates genome compartmentalization and loop extrusion barriers" (NSMB-A46086-T). Please carefully follow the step-by-step instructions provided in the attached file, and add a response in each row of the table to indicate the changes that you have made. Ensuring that each point is addressed will help to ensure that your revised manuscript can be swiftly handed over to our production team.

In recognition of the time and expertise our reviewers provide to Nature Structural & Molecular Biology's editorial process, we would like to formally acknowledge their contribution to the external peer review of your manuscript entitled "Heterochromatin diversity modulates genome compartmentalization and loop extrusion barriers". For those reviewers who give their assent, we will be publishing their names alongside the published article.

Nature Structural & Molecular Biology offers a Transparent Peer Review option for new original research manuscripts submitted after December 1st, 2019. As part of this initiative, we encourage our authors to support increased transparency into the peer review process by agreeing to have the reviewer comments, author rebuttal letters, and editorial decision letters published as a Supplementary item. When you submit your final files please clearly state in your cover letter whether or not you would like to participate in this initiative. Please note that failure to state your preference will result in delays in accepting your manuscript for publication.

Cover suggestions

As you prepare your final files we encourage you to consider whether you have any images or illustrations that may be appropriate for use on the cover of Nature Structural & Molecular Biology.

Nature Structural & Molecular Biology has now transitioned to a unified Rights Collection system which will allow our Author Services team to quickly and easily collect the rights and permissions required to publish your work. Approximately 10 days after your paper is formally accepted, you will receive an email in providing you with a link to complete the grant of rights. If your paper is eligible for Open Access, our Author Services team will also be in touch regarding any additional information that may be required to arrange payment for your article.

Please note that *Nature Structural & Molecular Biology* is a Transformative Journal (TJ). Authors may publish their research with us through the traditional subscription access route or make their paper immediately open access through payment of an article-processing charge (APC). Authors will not be required to make a final decision about access to their article until it has been accepted. [Find out more about Transformative Journals](https://www.springernature.com/gp/open-research/transformative-journals)

Authors may need to take specific actions to achieve [compliance with funder and institutional open access mandates](https://www.springernature.com/gp/open-research/funding/policy-compliance-faqs). If your research is supported by a funder that requires immediate open access (e.g. according to [Plan S principles](https://www.springernature.com/gp/open-research/plan-s-compliance)) then you should select the gold OA route, and we will direct you to the compliant route where possible. For authors selecting the subscription publication route, the journal's standard licensing terms will need to be accepted, including [self-archiving policies](https://www.springernature.com/gp/open-research/policies/journal-policies). Those licensing terms will supersede any other terms that the author or any third party may assert apply to any version of the manuscript.

Please use the following link for uploading these materials:
[Redacted]

Best regards,

Sophia Frank
Editorial Assistant
Nature Structural & Molecular Biology
nsmb@us.nature.com

On behalf of

Carolina Perdigoto, PhD
Chief Editor
Nature Structural & Molecular Biology
orcid.org/0000-0002-5783-7106

Reviewer #1:

Remarks to the Author:

The revised version of the manuscript has improved clarity and addresses many of my previous comments. I agree that some of the experiments I originally suggested would need a stretch of work that might be kept for future work and I am in favour of publication of the current version of the manuscript.

Reviewer #2:

Remarks to the Author:

The authors have addressed all the points that I raised and I don't have further concerns about the work.

Reviewer #3:

Remarks to the Author:

The authors have responded to the remarks this reviewer has raised for the first submission of this manuscript [Redacted]. They either revising the manuscript or convincingly discussing why certain analyses were not possible.

I find it acceptable for publication in Nature Structural & Molecular Biology, which is in my opinion a very suitable journal for this manuscript.

Final Decision Letter:

Message 1st Nov 2022

:

Dear Dr. Dekker,

We are now happy to accept your revised paper "Diverse silent chromatin states modulate

genome compartmentalization and loop extrusion barriers" for publication as a Article in Nature Structural & Molecular Biology.

Your paper will be published online soon after we receive proof corrections and will appear in print in the next available issue. You can find out your date of online publication by contacting the production team shortly after sending your proof corrections. Content is published online weekly on Mondays and Thursdays, and the embargo is set at 16:00 London time (GMT)/11:00 am US Eastern time (EST) on the day of publication. Now is the time to inform your Public Relations or Press Office about your paper, as they might be interested in promoting its publication. This will allow them time to prepare an accurate and satisfactory press release. Include your manuscript tracking number (NSMB-A46086A) and our journal name, which they will need when they contact our press office.

About one week before your paper is published online, we shall be distributing a press

release to news organizations worldwide, which may very well include details of your work. We are happy for your institution or funding agency to prepare its own press release, but it must mention the embargo date and Nature Structural & Molecular Biology. If you or your Press Office have any enquiries in the meantime, please contact press@nature.com.

Please note that *Nature Structural & Molecular Biology* is a Transformative Journal (TJ). Authors may publish their research with us through the traditional subscription access route or make their paper immediately open access through payment of an article-processing charge (APC). Authors will not be required to make a final decision about access to their article until it has been accepted. Find out more about Transformative Journals <https://www.springernature.com/gp/open-research/transformative-journals>

Authors may need to take specific actions to achieve [compliance with funder and institutional open access mandates](https://www.springernature.com/gp/open-research/funding/policy-compliance-faqs). If your research is supported by a funder that requires immediate open access (e.g. according to [Plan S principles](https://www.springernature.com/gp/open-research/plan-s-compliance)) then you should select the gold OA route, and we will direct you to the compliant route where possible. For authors selecting the subscription publication route, the journal's standard licensing terms will need to be accepted, including [self-archiving policies](https://www.springernature.com/gp/open-research/policies/journal-policies). Those licensing terms will supersede any other terms that the author or any third party may assert apply to any version of the manuscript.

Kind regards,
Florian

Dr Florian Ullrich
Associate Editor, Nature
Consulting Editor, Nature Structural & Molecular Biology
ORCID 0000-0002-1153-2040